# Petrogenesis of the Eudialyte Complex of the Lovozero Alkaline Massif (Kola Peninsula, Russia)

**Julia A. Mikhailova \*, Gregory Yu. Ivanyuk, Andrey O. Kalashnikov** 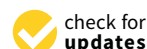**, Yakov A. Pakhomovsky, Ayya V. Bazai and Victor N. Yakovenchuk**

Kola Science Centre, Russian Academy of Sciences, 14 Fersman Street, Apatity 184209, Russia;
g.ivanyuk@gmail.com (G.Y.I.); kalashnikov@geoksc.apatity.ru (A.O.K.); pakhom@geoksc.apatity.ru (Y.A.P.);
bazai@geoksc.apatity.ru (A.V.B.); yakovenchuk@geoksc.apatity.ru (V.N.Y.)
**\*** Correspondence: ylya_korchak@mail.ru; Tel.: +7-81555-79333

**Abstract:** The Lovozero Alkaline Massif intruded through the Archaean granite-gneiss and Devonian volcaniclastic rocks about 360 million years ago, and formed a large (20 × 30 km) laccolith-type body, rhythmically layered in its lower part (the Layered Complex) and indistinctly layered and enriched in eudialyte-group minerals in its upper part (the Eudialyte Complex). The Eudialyte Complex is composed of two groups of rocks. Among the hypersolvus meso-melanocratic alkaline rocks (mainly malignite, as well as shonkinite, melteigite, and ijolite enriched with the eudialyte-group minerals, EGM), there are lenses of subsolvus leucocratic rocks (foyaite, fine-grained nepheline syenite, urtite with phosphorus mineralization, and primary lovozerite-group minerals). Leucocratic rocks were formed in the process of the fractional crystallization of melanocratic melt enriched in Fe, high field strength elements (*HFSE*), and halogens. The fractionation of the melanocratic melt proceeded in the direction of an enrichment in nepheline and a decrease in the aegirine content. A similar fractionation path occurs in the $Na_2O$-$Al_2O_3$-$Fe_2O_3$-$SiO_2$ system, where the melt of the "ijolite" type (approximately 50% of aegirine) evolves towards "phonolitic eutectic" (approximately 10% of aegirine). The temperature of the crystallization of subsolvus leucocratic rocks was about 550 °C. Hypersolvus meso-melanocratic rocks were formed at temperatures of 700–350 °C, with a gradual transition from an almost anhydrous *HFSE*-Fe-Cl/F-rich alkaline melt to a Na(Cl, F)-rich water solution. Devonian volcaniclastic rocks underwent metasomatic treatment of varying intensity and survived in the Eudialyte Complex, some remaining unchanged and some turning into nepheline syenites. In these rocks, there are signs of a gradual increase in the intensity of alkaline metasomatism, including a wide variety of zirconium phases. The relatively high fugacity of fluorine favored an early formation of zircon in apo-basalt metasomatites. The ensuing crystallization of aegirine in the metasomatites led to an increase in alkali content relative to silicon and parakeldyshite formation. After that, EGM was formed, under the influence of Ca-rich solutions produced by basalt fenitization.

**Keywords:** Lovozero Alkaline Massif; fractional crystallization; fenitization; nepheline; alkali feldspar; clinopyroxene; amphibole; eudialyte

## 1. Introduction

Alkaline igneous rocks are among the rarest magmatic rocks. These rocks contain either (1) modal feldspathoids or alkali amphiboles or pyroxenes or (2) normative feldspathoids or aegirine [1]. Based on the molar ratios of $Na_2O$, $K_2O$, and $CaO$ relative to $Al_2O_3$, these rocks can be subdivided into metaluminous {$(Na_2O + K_2O) < Al_2O_3 < (Na_2O + K_2O + CaO)$}, peraluminous {$Al_2O_3 > (Na_2O + K_2O + CaO)$} and peralkaline {$(Na_2O + K_2O) > Al_2O_3$} types [2].

Alkaline rocks are extraordinary rich in large ion lithophile elements (*LILE*), such as Na and K, and in high field strength elements (*HFSE*), such as Ti, Zr, Hf, Nb, Ta, rare-earth elements (*REE*), U and Th, forming economically important deposits of these elements [3,4]. Primary magmatic minerals of *HFSE* in alkaline rocks form two main mineral associations. Minerals with relatively simple crystal structures (in terms of topological complexity of crystal structures [5]), such as zircon/baddeleyite and titanite/perovskite, ilmenite and titanomagnetite (depending on the $SiO_2$ activity), which belong to the miaskite association [3]. Complex Na-Ca-*HFSE-REE* silicates (e.g., minerals of the eudialyte group, rinkite, aenigmatite, lamprophyllite, astrophyllite, etc.) form agpaitic assemblages. Hyperagpaitic assemblages include ussingite, naujakasite, steenstrupine-(Ce), members of the lovozerite and lomonosovite groups, and partly water-soluble minerals (e.g., villiaumite, natrosilite, natrophosphate, and thermonatrite). Transitional agpaitic rocks contain *HFSE* minerals that are typical of all three types of alkaline rocks (for example, titanite and eudialyte or eudialyte and lovozerite).

The field occurrence of agpaitic rocks is variable. The Lovozero Alkaline Massif, together with the nearby Khibiny Massif and the Ilímaussaq Complex of Greenland, is one of the sites with the world's largest occurrences of agpaitic nepheline syenite intrusions. The causes of the enrichment of agpaitic rocks in *HFSE* and *REE* are studied in detail. Agpaitic nepheline syenites form by extensive differentiation of parental mafic magmas at low oxygen fugacity [6]. Such conditions determine the presence of $CH_4$-rich fluids [7–9] instead of $H_2O$–$CO_2$ fluid mixtures typical of less reduced rock types. Since Na, Cl, and F are well soluble in water, these elements do not pass into the fluid, but remain in the melt [10,11]. In addition, the chlorine fluid/melt partition coefficient decreases with growth of fluorine content, and vice versa [10]. As *HFSE* and *REE* have high solubilities in Na-, Cl- and F-rich melts [12,13], they will eventually form agpaitic minerals such as the eudialyte-group minerals (EGM), rinkite, aenigmatite, (baryto)lamprophyllite and astrophyllite. Agpaitic rocks can be extremely rich in EGM; for example, the upper part of the Lovozero Massif is composed of alkaline rocks with rock-forming EGM (up to 90 mod. %), due to which it received the name of "the Eudialyte Complex" [14]. The EGM-rich rocks are represented here by foid syenite, foidolite, and alkaline metasomatite and pegmatite. In this article, based on the petrography of these rocks and data on the chemical composition of rock-forming minerals, we present estimations of conditions and mechanisms of the Lovozero "Eudialyte Complex" formation.

## 2. Geological Setting

The laccolith-type Lovozero Alkaline Massif was emplaced 360–370 million years ago [15–17] into Archean granite gneisses covered by Devonian volcaniclastic rocks [18]. According to geophysical studies [19], alkaline rocks are traced to a depth of 7 km, the lower limit of their distribution is not detected. The laccolith has a size 20 × 30 km on the day surface, and about 12 × 16 km on a 5 km depth [14]. In the upper part, the intrusion contacts with host rocks are almost vertical. On the plane, the massif has the shape of a quadrilateral with rounded corners (Figure 1a).

The Lovozero Massif is a layered intrusion composed of two macro units: the Eudialyte Complex and the Layered Complex. The Eudialyte Complex is located at the top of the massif, accounts for 18% of its volume and is not layered. The Layered Complex, with the layering clearly manifested, occupies 77% of the volume of the massif. The elementary unit of layering here is a sequence ("rhythm" or "pack") of alkaline rocks (from the bottom up): urtite–foyaite–lujavrite (Figure 1b,c). In this series, there is a gradual transition from almost monomineral nepheline or nepheline-kalsilite foidolite (urtite) to leucocratic nepheline (±sodalite) syenite (foyaite), and then to lujavrite (meso- and melanocratic nepheline syenite of the malignite–shonkinite rock series). The textures of these rocks change from massive in urtite to semi-trachytoid in foyaiteand trachytoid in lujavrite due to the gradual ordering of the feldspar plates. The sequence urtite–foyaite–lujavrite is repeated regularly [14,20]. The contact between the underlying lujavrite and the overlying urtite is sharp and marked by rich loparite impregnation in both nepheline syenite and foidolite (50–80 cm thick loparite maligite–ijolite ore-horizons [21,22]). The thickness of the individual rhythm is 5 to 100 m-they lie

sub-horizontally (the dip is 5–15° towards the center of the massif) and have nearly uniform thickness. The rhythms of the Layered Complex are combined into seven series, denoted by Roman numerals, and the series, in turn, are grouped into three zones. The upper zone (up to 370 m) consists of packs of urtite–foyaite–lujavrite. The middle zone, with a thickness of 640–670 m, is composed of monotonous lujavrite with sparse interlayers of foyaite. The lower zone consists predominantly of foyaite-lujavrite packs [14]. The boundaries of the zones are the urtite horizons II (series) -5 (rhythm) and III-1 (Figure 1c).

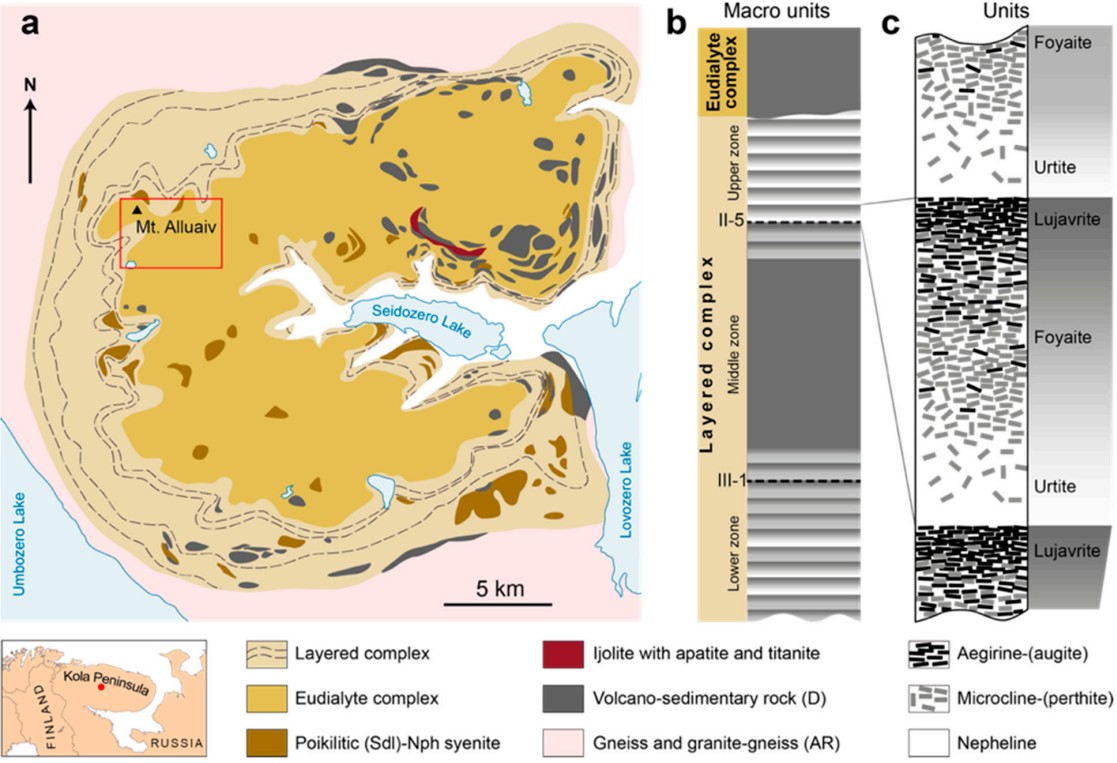

**Figure 1.** The geology of the Lovozero Alkaline Massif: schematic geological map, after [20] (**a**) and stratigraphic column [14] (**b**), with principal scheme of layering [21] (**c**).

Other rock types of the Lovozero Massif are subordinate to their layering and form sub-horizontal layers or lenses in the Eudialyte Complex and the Layered Complex (Figure 1a). Xenoliths of volcaniclastic rocks are ubiquitous. Unchanged xenoliths are composed of interbedded olivine basalts, basalt tuffs, tuffites, sandstones, and quartzites. However, usually these lithologies are deeply metasomatized [18]. Numerous lenses of poikilitic nepheline and sodalite-nepheline (sometimes with nosean) syenite, as well as alkaline pegmatites and hydrothermal veins, are located within the eudialyte and layered complexes [23–25].

The thickness of the Eudialyte Complex in different parts of the massif ranges from 100 to 800 m. According to [14,20], Zr-rich melts break through and overlap the previously formed Layered Complex. The cutting of the upper rhythms of the urtite–foyaite–lujavrite is visible; the plane of the contact of the complexes falls to the center of the massif, with the angle of dip increasing in the same direction from 10 to 40°. In fact, the Eudialyte Complex can be regarded as part of the giant Lovozero Eudialyte Deposit that includes several rich sites, in particular, the Karnasurt, Kedykvyrpakhk, Alluaiv, Angvundaschorr, Sengischorr and Parguaiv sites, and the Alluaiv site is the best explored of them [4].

## 3. Materials and Methods

As the Alluaiv site (Figure 2) is the best explored, it was chosen for detailed study of the Eudialyte Complex, whose thickness varies here from 280 to 350 m. There is a wide diversity of EGM-rich

alkaline rocks–a dense drill grid and numerous outcrops display the rock relations and their contacts with the underlying rocks of the Layered Complex. As a whole, we used 275 thin polished sections of rocks selected from cores of 27 exploration boreholes and the day surface (Figure 2a).

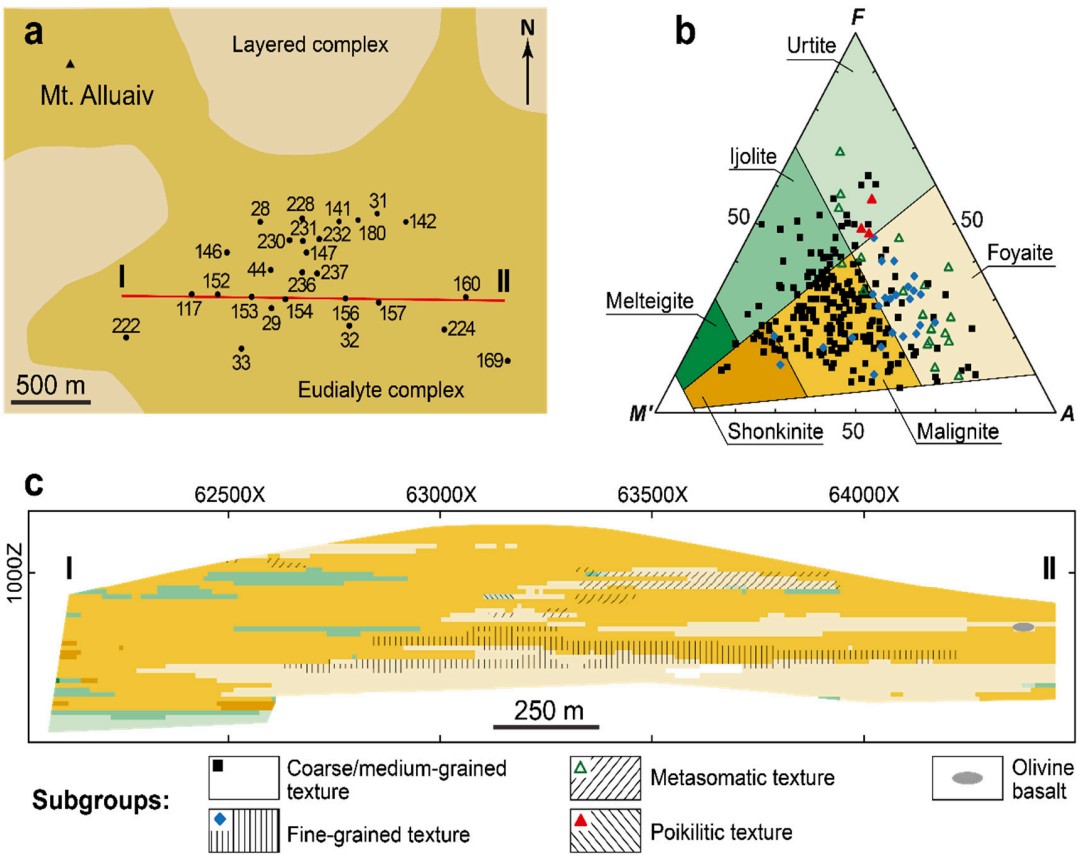

**Figure 2.** The Alluaiv site of the Lovozero Eudialyte Deposit (see Figure 1a): (**a**)—scheme of drill holes; (**b**)—the modal composition of alkaline rocks of the Alluaiv site. Rocks with fine-grained (blue diamonds), metasomatic (green triangles) and poikilitic (red triangles) textures are highlighted with special icons; (**c**)—section along the I–II line (see Figure 2a). The rock legend corresponds to Figure 2b. Rock with fine-grained, metasomatic, and poikilitic textures are highlighted by hatching.

The thin polished sections were analyzed using the scanning electron microscope LEO-1450 (Carl Zeiss Microscopy, Oberkochen, Germany), with energy-dispersive microanalyzer Röntek to obtain BSE (Back Scattered Electron) images and pre-analyze all detected minerals. The chemical composition of rock-forming minerals was analyzed with a Cameca MS-46 electron microprobe (Cameca, Gennevilliers, France) operating in WDS-mode at 22 kV with beam diameter 10 μm, beam current 20–40 nA and counting times 20 s (for a peak) and $2 \times 10$ s (for background before and after the peak), with 5–10 counts for every element in each point. The analytical precision (reproducibility) of mineral analyses is 0.2–0.05 wt. % (2 standard deviations) for the major element and about 0.01 wt. % for impurities. The standards used, the detection limits, and the analytical accuracy values are given in Table S1 in Supplementary Materials. The systematic errors were within the random errors.

Major elements in rocks were determined by a wet chemical analysis in the Geological Institute of KSC RAS. The accuracy limits of the wet chemical analysis are given in Table S1 (Supplementary Materials). Cation contents were calculated using the MINAL program of D. Dolivo-Dobrovolsky [26]. The amphibole-group mineral formulae were calculated based on O + OH + F = 24 atoms per formula unit and OH = 2 – 2Ti. The formula calculation was performed following the IMA 2012 recommendations [27] using the Excel spreadsheet of Locock [28]. Statistical analyses were carried out using the STATISTICA 13 [29] and TableCurve 2.0 [30] programs. For the statistics, resulting

values of the analyses below the limit of accuracy (see Table S1 in Supplementary Materials) were considered ten times lower than the limit. Geostatisical studies and 3D modeling were conducted with the MICROMINE 16.1 program (Micromine Pty Ltd., Perth, Australia). Interpolation was performed by ordinary kriging. The ImageJ program (US National Institutes of Health) was used to generate digital images from the thin polished sections images and determinate modal proportion of rock-forming minerals.

The mineral abbreviation used include Aeg—aegirine-(augite), Ab—albite, All—alluaivite, Ap—fluorapatite, Aq—aqualite, Di—diopside, Eud—minerals of eudialyte group (EGM), Fo—forsterite, Ilm—ilmenite, Kap—kapustinite, Kent—kentbrooksite, Ks—kalsilite, Lmp—lamprophillite, Lom—lomonosovite, Lop—loparite, Mag—magnetite, Marf—magnesioarfvedsonite, Mc—microcline (-perthite), Nph—nepheline, Ntr—natrolite, On—oneillite, Or—orthoclase, Phl—phlogopite, Pkl—parakeldyshite, Prv—perovskite, Ras—raslakite, Rct—richterite, Sdl—sodalite, Sp—sphalerite, Tas—taseqite, Ttn—titanite, Umb—umbozerite, Zir—zirsilite.

## 4. Results

### 4.1. Petrography

The basis of the petrographic classification of plutonic alkaline rocks of the Alluaiv site is the QAPF (Q—quartz, A—alkali feldspar, P—plagioclase, F—feldspathoid) classification of the International Union of Geological Sciences [1], which takes into account the ratio of K-Na feldspar (*A*), feldspathoids (*F*) and mafic minerals (color index *M'*) in the rock (Table S2 in Supplementary Materials). Within the *AFM'* triangle, the rock points are located mainly in the center, do not form isolated groups and are divided into foid syenite (shonkinite, malignite and foyaite) and foidolite (melteigite, ijolite and urtite) (Figure 2b).

The most common rocks of the Alluaiv site are coarse- to medium-grained trachytoid malignites (Figure 2b,c and Figure 3a). In these rocks, euhedral plates of microcline-perthite are oriented subparallel one to another. The feldspar crystals have the most intensely developed faces of pinacoid (010), are elongated along [001] up to 3 cm, with the average side ratio *a:b:c* = 6:1:10 (data on 82 crystals). Microcline-perthite crystals are often slightly bent and fractured. Albites concentrate usually around the fractures (Figure 3b), making them well visible under the microscope. In addition, small albite perthites are evenly distributed within microcline crystals. In sections perpendicular to the pinacoid (010), albite occupies 5% to 37% (median is 17%) of the microcline-perthite grain. As a rule, the albite is intensively replaced by natrolite (Figure 3b,d).

Feldspathoids fill interstices between the microcline-perthite plates. The most common of them, nepheline, forms grains of two morphological types. Nepheline-I occurs as euhedral crystals with numerous small inclusions of aegirine and microcline (Figure 3a,c). Aegirine inclusions are presented by randomly oriented short-prismatic crystals (up to 200 μm long), long-prismatic crystals (up to 600 μm long) oriented parallel to nepheline grain faces, as well as irregular intergrowths of small needles (on average 3 × 8 μm). Nepheline-II forms lens-like anhedral grains with small inclusions of aegirine-(augite) on the periphery (Figure 3b). Sodalite and natrolite, except for pseudomorphs after nepheline (Figure 3c) and albite (Figure 3d), form primary grains, similar in morphology to nepheline-II and coexisting with it. Inside the grains of primary sodalite, natrolite or natrolite–sodalite aggregates occur (Figure 3d). Primary natrolite often contains numerous small (up to 20 μm) boehmite inclusions.

The feldspar and feldspathoid grains are surrounded by "streams" of subparallel-oriented crystals of mafic minerals (Figure 3a–f), represented by alkaline clinopyroxenes (aegirine and aegirine-augite) and amphiboles (mainly magnesioarfvedsonite), as well as lamprophyllite. The marginal parts of the "streams" consist of small (50 × 200 μm in the average) long-prismatic clinopyroxene crystals, and the axial zones is composed of larger anhedral amphibole grains (Figure 3b–d). The edge parts of the amphibole crystals usually contain clinopyroxene inclusions oriented according to the general

direction of the "stream". Lamprophyllite forms either poikilitic crystals (Figure 3c) or individual plates (Figure 3d) located between grains of clinopyroxenes and amphiboles.

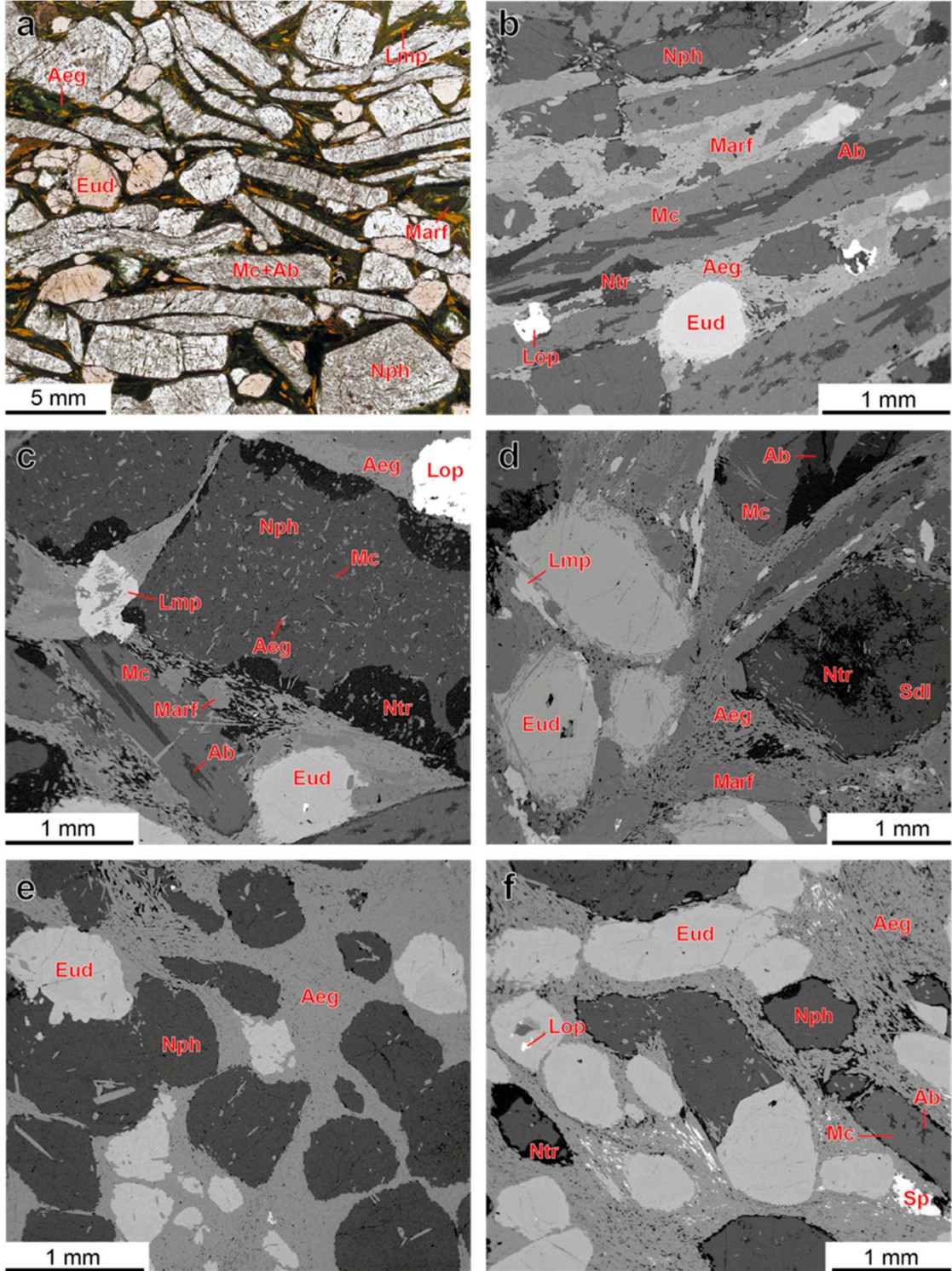

**Figure 3.** Meso-melanocratic rocks of the Alluaiv site: (**a**)—EGM-rich trachytoid malignite (lujavrite) 117/76 (photo of polished thin section in transmitted light); (**b**–**d**)—BSE images of EGM-rich malignites (lujavrites) 117/121 (**b**), 154/85 (**c**) and 29/37 (**d**); (**e**)—BSE-image of EGM-rich ijolite 222/177; (**f**)—BSE-image of shonkinite 32/42.4.

Interstices between grains of above-mentioned mafic minerals of the "streams" are filled with natrolite. If the proportion of natrolite is large, clinopyroxenes and amphiboles form euhedral short-prismatic crystals (Figure 3c). Similar to amphiboles, EGM grains are in the axial parts of the "streams". They usually form rounded or lenticular grains (in the average, 2.3 mm in diameter), while their euhedral crystals (up to 7 mm long) with rhombic prismatic and rhombohedral faces occur much more rarely. The marginal parts (up to 200 µm) of EGM grains are saturated with inclusions of aegirine-(augite) and, in smaller quantities, magnesioarfvedsonite. The inclusions are oriented according to the general direction of the "stream" (Figure 3b–d). The length of common boundaries between EGM and clinopyroxene + amphibole grains significantly exceeds that between EGM and leucocratic minerals (9:1 ratio).

The accessory minerals of malignite are in the interstices between clinopyroxene and amphibole crystals. The most common (found in more than 50% of samples) accessory minerals include the perovskite group minerals (loparite, lueshite), sulfides (sphalerite, pyrrhotite, chalcopyrite), stronadelphite, pyrochlore group minerals, baritolamprophillite, chlorbartonite, thorite, and thorianite. Arsenopyrite, löllingite, cobaltine, lorenzenite, and rinkite are less common. Secondary minerals are presented by lovozerite (after EGM), rhabdophane-(La/Ce), bastnaesite-(Ce), strontianite, barite, witherite, hanneshite, and cerussite.

With an increase of feldspathoid content at the expense of microcline-perthite, the malignite transits to ijolite (Figure 3e). The trachytoid texture is lost in this case, but the "streams" of mafic minerals remain, which are now oriented arbitrarily. In this rock, there are also two morphological types of nepheline, primary natrolite and primary sodalite with natrolite cores. Again, EGM form here rounded, oval, or irregularly shaped grains with aegirine-(augite) inclusions on the periphery. Less widespread are well-formed EGM crystals and their poikilitic grains within aegirine-(augite) segregations.

An increase in the total content of mafic minerals in malignite, with its formal transition to shonkinite (Figures 2b and 3f), results from an increase in the EGM proportion. At the same time, the content of all mafic minerals excluding EGM (i.e., M' minus EGM, modal %) never exceeds 34–38 modal %. The average EGM content in shonkinites is 22.4 modal %. In fact, there are no differences between malignite and shonkinite besides EGM contents.

With a decrease in total content of mafic minerals (including EGM), mesocratic rocks (malignite and ijolite) turn into foyaite or, with an increase in the proportion of feldspathoids, into urtite (Figure 4a). Both these transitions are gradual, occur in narrow (1–3 cm) intervals and result in rock texture isotropization. The next important change is the appearance of primary albite (Figure 4a), which increases the total content of this mineral to 29 modal %.

In foyaite and urtite, mafic minerals do not form a communicating system of "streams": fine-grained aggregates of prismatic aegirine-(augite) and rounded EGM grains fill interstices between microcline-perthite, albite and nepheline grains (Figure 4a), while amphiboles form poikilitic crystals. When the color index M' of the rock decreases (apart from amphiboles) first clinopyroxenes and then EGM form poikilitic crystals. Nepheline forms euhedral crystals with or without inclusions of microcline and aegirine. Natrolite replaces nepheline also forms primary grains in association with unchanged nepheline. Primary sodalite is present too. The set of accessory minerals in foyaite and urtite is similar to that in malignite, but phosphorus-rich minerals are much more widely spread in the leucocratic rocks. Stronadelphite, fluorapatite, lomonosovite (Figure 4b), vuonnemite, bornemanite, monazite-(Ce), xenotime-(Y), nastrophite and nabaphite are characteristic of these rocks as well as lovozerite-group minerals (lovozerite, kapustinite, litvinskite). The latter not only replace EGM, but also occur as primary minerals (Figure 4d).

Fine-grained nepheline syenite constitutes a significant part of the studied samples (Figure 2b). The basis of their texture is formed by euhedral albite crystals (up to 32 modal %; in the average, 25 modal %). Between the albite crystals, anhedral grains of microcline (without perthites) and nepheline occur. Aegirine(-augite) forms prismatic crystals, which are either uniformly dispersed

in the rock or form small lenses in aggregates of leucocratic minerals. Rounded grains of EGM are uniformly disseminated in the rock (Figure 4c,d).

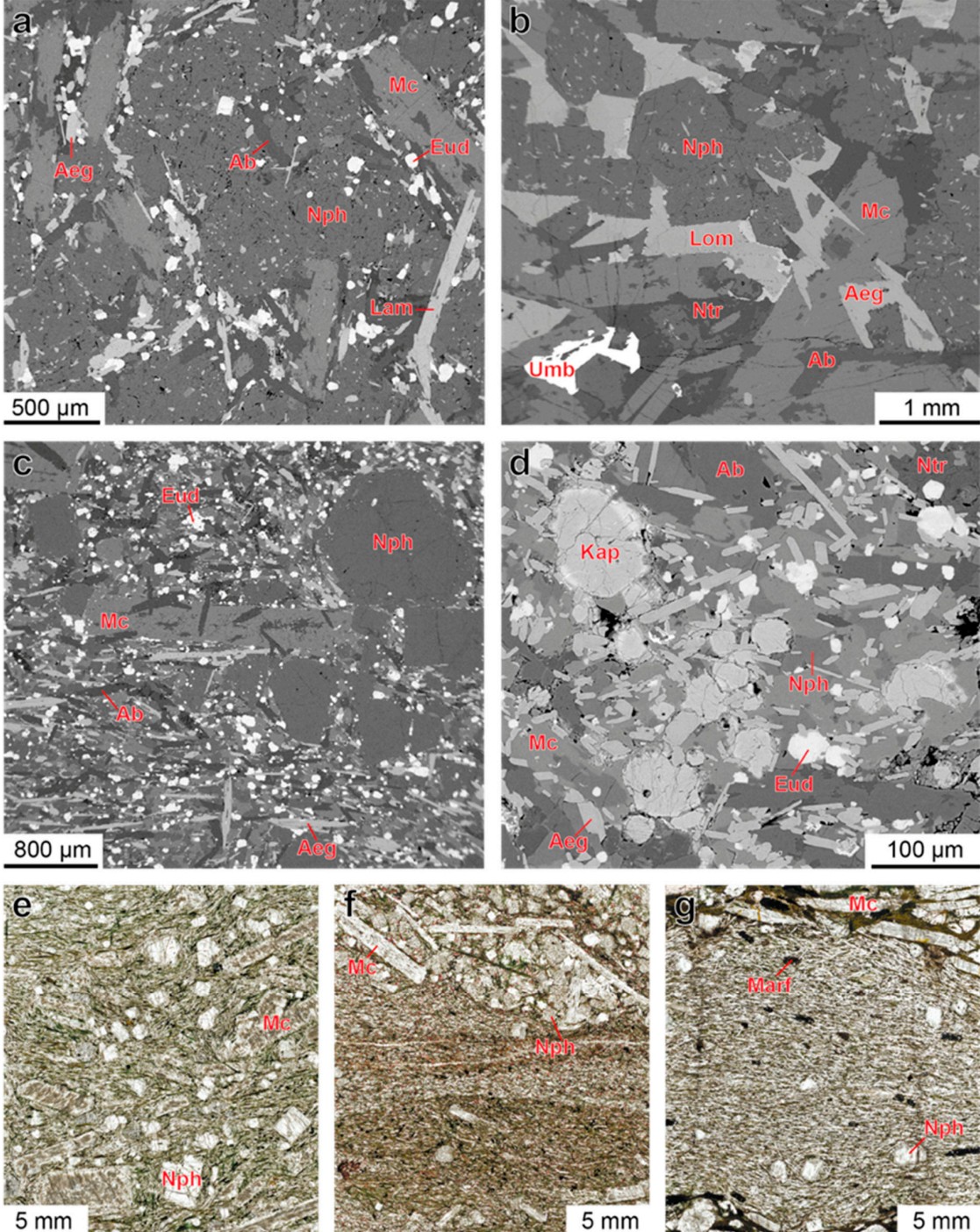

**Figure 4.** Leucocratic rocks of the Alluaiv site: (**a**)—BSE-image of urtite 117/208; (**b**)—BSE-image of foyaite 154/300; (**c**)—BSE-image of fine-grained foyaite 157/36; (**d**)—BSE-image of kapustinite-rich fine-grained foyaite 32/243; (**e**,**f**)—images in transmitted light of polished thin sections of porphyry foyaite 154/253 (**e**); contact of fine-grained foyaite (below) and urtite 154/216 (**f**); vein of fine-grained foyaite in malignite 157/36 (**g**).

In terms of modal composition, most samples of fine-grained nepheline syenite correspond to foyaite (Figure 2b); the transition to malignite and shonkinite occurs due to the appearance of poikilitic crystals of (magnesio)arfvedsonite (Figure 4g), murmanite, lomonosovite, the lovozerite-group minerals, and lamprophyllite. Microcline-perthite and nepheline form large phenocrysts (content up to 40 modal %, size up to 1 cm) in fine-grained mass (Figure 4c,e–g). In the narrow (up to 100 μm) marginal zone of the phenocrysts, there are small inclusions of aegirine(-augite) and EGM oriented usually subparallel to the phenocryst faces (Figure 4c). With an increase in the content of microcline-perthite and/or nepheline phenocrysts, fine-grained nepheline syenite transforms into medium- to coarse-grained foyaite or urtite (Figure 4f). It should be noted that the co-existence of primary lovozerite-group minerals and unchanged EGM is characteristic of fine-grained nepheline syenite. Lovozerite-group minerals form either small rounded grains (Figure 4d) or large poikilitic crystals.

Usually, the contact between fine-grained nepheline syenite and malignite is sharp and concordant with orientation of microcline plates in the malignite, i.e., trachytoid plane. In fine-grained rock, long-prismatic clinopyroxene crystals and albite plates are also co-oriented with the contact (Figure 4g). Fine-grained nepheline syenite form various lens-like bodies (5 mm to 100 m thick, Figures 2c and 4g) among the coarse- and medium-grained rocks of the Eudialyte Complex.

Comparatively fresh olivine basalt is a rare rock of the Alluaiv site (Figures 2c and 5a,b). In this rock, forsterite crystals (up to 1.3 mm wide), phlogopite (up to 0.6 mm in diameter) and diopside/augite (up to 0.8 mm wide) are disseminated into the main fine-grained diopside (augite)-phlogopite mass (the average size of individual grains is about 30 μm). Often, diopside (augite) grains are resorbed by phlogopite, and forsterite is serpentinized. Magnetite forms skeletal crystals and small (8 μm on the average) rounded grains, perovskite constitutes chains of cubic crystals, and fluorite, fluorapatite and pectolite are located within phlogopite-diopside (augite) aggregate and inside the skeletal magnetite crystals. Accessory minerals are barite, djerfisherite, and pyrrhotite. High-temperature fenitization of olivine basalt and its tuff produced numerous metasomatic rocks [18], up to formation of fine- to medium-grained nepheline syenites with relics of earlier minerals and metasomatic textures.

Rocks that modally correspond to foyaite and urtite but have a metasomatic texture, are highlighted in Figure 2b separately. They form inclined layers (Figure 2c) and small individual lenses in the Eudialyte Complex and consist mainly of large orthoclase crystals, which can be both homogeneous and perthite-bearing. Orthoclase is intensively resorbed by nepheline that often includes small irregularly shaped orthoclase relics (Figure 5c). Moreover, nepheline forms large (up to 1 cm) euhedral crystals that contain magnesioarfvedsonite inclusions and symplectic intergrowths (Figure 5c,d) and are rimmed by aegirine. Also, in this rock, there is a large number of resorbed augite relics surrounded by zonal rims of richterite, ferri-katophorite and eckermannite), then aegirine-augite and marginal aegirine. In addition to these rims, aegirine occurs as fan-shaped aggregates among orthoclase-perthite and nepheline. Characteristic accessory minerals located mainly among aegirine include parakeldyshite, titanite, ilmenite, zircon, baddeleyite, fluorapatite, and EGM. These minerals usually form various zonal segregations (Figure 5d): zircon + baddeleyite (inside)–parakeldyshite (outside), parakeldyshite (inside)–EGM (outside), titanite + fluorapatite (inside)–EGM (outside); ilmenite (inside)–titanite (outside).

The rocks with poikilitic texture (Figure 2b) correspond modally to feldspar-bearing urtite consisting of large (up to 8 cm in diameter) orthoclase crystals with numerous inclusions of sodalite, natrolite and nepheline (Figure 5e,f). Aegirine(-augite) is the main mafic mineral, but its content does not exceed 18 modal %. EGM form rare poikilitic crystals with inclusions of surrounding minerals. Such rocks are rare (1% of the studied samples) and were found only in one drill hole (154, interval 999–931 m).

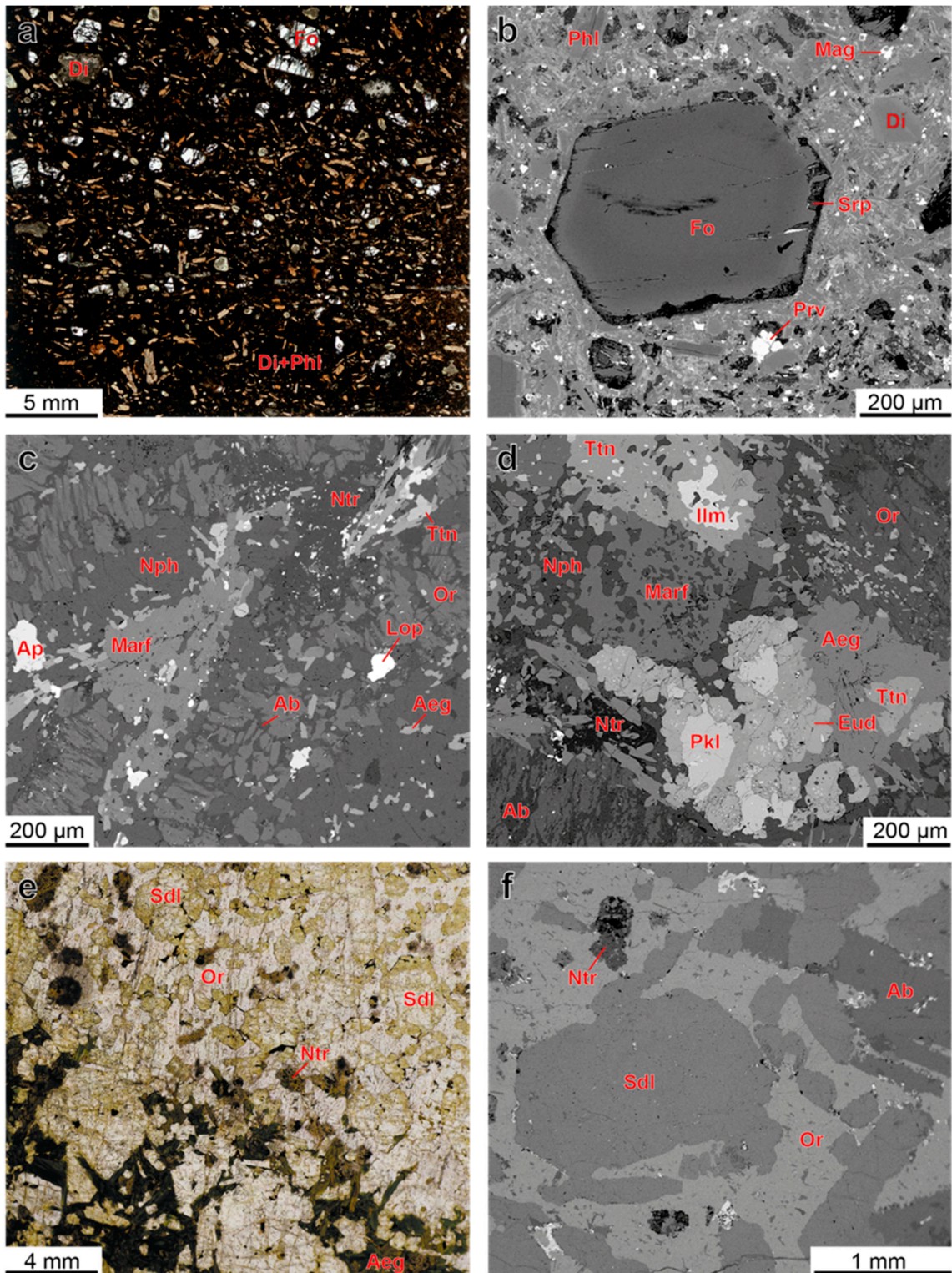

**Figure 5.** Olivine basalt and products of its deep metasomatic alteration (**a–d**): photo in transmitted light (**a**) and BSE-image (**b**) of polished thin section of olivine basalt 160/130; orthoclase-rich metasomatic rocks 156/66 (**c**) and 156/98 (**d**). Poikilitic urtite 154/153: photo in transmitted light (**e**) and BSE-image (**f**).

### 4.2. Whole-Rock Chemistry

Data on the concentrations of petrogenic elements in rocks of the Lovozero Eudialyte Complex are shown in Figure 6 and in Table S3 (Supplementary Materials). The most important changes in

the chemical composition of rocks occur with an increase in the proportion of mafic minerals in their composition. During the transitions from foyaite to malignite and from urtite to ijolite, when the color index $M'$ exceeds 30% (Figure 2b), content of $Al_2O_3$ in the rock composition gradually decreases, while the contents of $Na_2O$ and $K_2O$ do not change. This leads to the predominance of the sum of these alkalis over aluminum. The proportion of samples with $K_a = (Na_2O + K_2O)/Al_2O_3 > 1$ (agpaitic coefficient) among the malignite, ijolite and shonkinite is 48%. As concentrations of $ZrO_2$, $TiO_2$, $Fe_2O_3$, MnO, MgO and SrO increase with $K_a$ growth (Figure 7) at the expense of $P_2O_5$, the more melanocratic rocks are EGM-richer and phosphate-poorer.

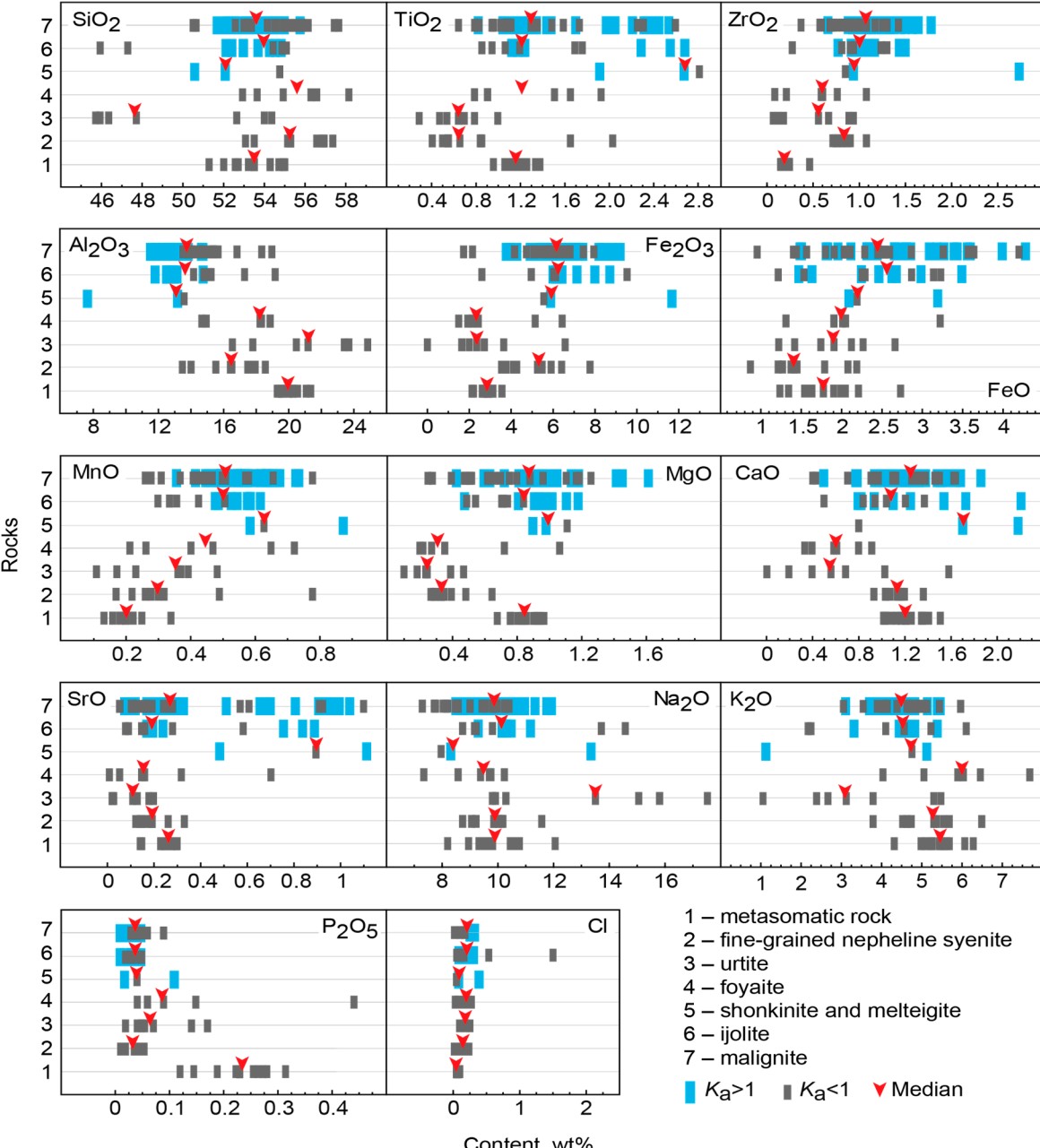

**Figure 6.** Variations in the chemical composition of rocks of the Alluaiv site. The rocks are divided into two groups by the value $K_a = (Na_2O + K_2O)/Al_2O_3$. Red arrows show the median contents.

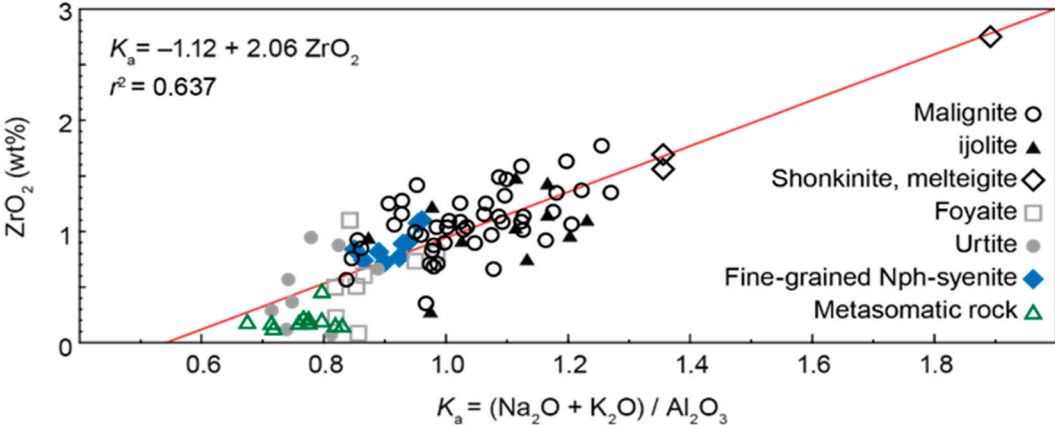

**Figure 7.** Content of $ZrO_2$ in rocks as a function of their agpaitic coefficient, $K_a$.

The composition of fine-grained nepheline syenite is generally similar to that of foyaites, except for CaO and $Fe_2O_3$, whose content in fine-grained rocks is higher. Urtite is relatively rich in $Al_2O_3$, $Na_2O$, and Cl, but it contains the smallest amounts of divalent cations, $TiO_2$, $K_2O$, and $SiO_2$. Metasomatic rocks have the highest concentrations of $P_2O_5$; their CaO and MgO contents are the highest among leucocratic rocks (Figure 6).

### 4.3. Mineral Chemistry

#### 4.3.1. Alkali Feldspars

Representative analyses of alkali feldspars are presented in Table 1, and all analyses are in Table S4 (Supplementary Materials).

**Table 1.** Representative microprobe analyses of alkali feldspar, wt. %.

| Rock | Malignite | | Ijolite | | Shonkinite, Melteigite | | Foyaite | | Urtite | | Fine-Grained Nph-Syenite | | Metasomatic Rock | |
|---|---|---|---|---|---|---|---|---|---|---|---|---|---|---|
| Drill hole | 29 | 117 | 153 | 160 | 153 | 222 | 154 | 224 | 117 | 147 | 160 | 157 | 156 | 156 |
| Deep, m | 37.5 | 102 | 143 | 120 | 36 | 123 | 271 | 102 | 208 | 85 | 172 | 36 | 77 * | 77 ** |
| $SiO_2$ | 63.98 | 64.90 | 65.32 | 65.88 | 65.28 | 65.08 | 64.24 | 65.14 | 63.92 | 65.74 | 65.53 | 63.80 | 64.98 | 64.40 |
| $Al_2O_3$ | 16.95 | 18.04 | 18.32 | 18.43 | 17.34 | 18.03 | 18.38 | 18.33 | 17.88 | 18.18 | 17.61 | 17.54 | 19.62 | 19.15 |
| $Fe_2O_3$ | 0.05 | 0.06 | b.d. | 0.03 | b.d. | 0.09 | 0.02 | 0.04 | b.d. | 0.08 | 0.51 | 0.65 | 0.38 | 0.45 |
| BaO | 0.14 | b.d. | b.d. | b.d. | b.d. | 0.12 | b.d. | 0.08 | 0.13 | b.d. | 0.12 | b.d. | 1.60 | 1.32 |
| $Na_2O$ | 0.30 | 0.26 | 0.35 | 0.38 | 0.30 | 0.32 | 0.43 | 0.34 | 0.39 | 0.39 | 1.74 | 0.64 | 5.00 | 4.41 |
| $K_2O$ | 16.72 | 16.19 | 16.43 | 15.89 | 16.89 | 16.32 | 16.15 | 16.07 | 16.04 | 16.20 | 14.05 | 15.72 | 6.75 | 7.52 |
| Sum | 98.14 | 99.45 | 100.42 | 100.61 | 99.81 | 99.96 | 99.22 | 100.00 | 98.36 | 100.59 | 99.56 | 98.35 | 98.33 | 97.25 |
| *Formula based on 8 oxygen atoms, apfu* | | | | | | | | | | | | | | |
| Si | 3.03 | 3.01 | 3.01 | 3.01 | 3.03 | 3.01 | 2.99 | 3.01 | 3.01 | 3.02 | 3.02 | 3.00 | 2.97 | 2.98 |
| Al | 0.95 | 0.99 | 0.99 | 0.99 | 0.95 | 0.98 | 1.01 | 1.00 | 0.99 | 0.98 | 0.96 | 0.97 | 1.06 | 1.04 |
| $Fe^{3+}$ | - | - | - | - | - | - | - | - | - | - | 0.02 | 0.02 | 0.01 | 0.02 |
| Ba | - | - | - | - | - | - | - | - | - | - | - | - | 0.03 | 0.02 |
| Na | 0.03 | 0.02 | 0.03 | 0.03 | 0.03 | 0.03 | 0.04 | 0.03 | 0.04 | 0.03 | 0.16 | 0.06 | 0.44 | 0.39 |
| K | 1.01 | 0.96 | 0.97 | 0.93 | 1.00 | 0.96 | 0.96 | 0.95 | 0.96 | 0.95 | 0.83 | 0.94 | 0.39 | 0.44 |
| Sum | 5.02 | 4.98 | 5.00 | 4.96 | 5.01 | 4.98 | 5.00 | 4.99 | 5.00 | 4.98 | 4.99 | 4.99 | 4.90 | 4.89 |
| *Mol. % endmembers* | | | | | | | | | | | | | | |
| Or | 97 | 98 | 97 | 96 | 97 | 97 | 96 | 97 | 96 | 96 | 82 | 92 | 44 | 50 |
| Ab | 3 | 2 | 3 | 4 | 3 | 3 | 4 | 3 | 4 | 4 | 16 | 6 | 51 | 45 |
| Csn | - | - | - | - | - | - | - | - | - | - | - | - | 3 | 3 |
| For | - | - | - | - | - | - | - | - | - | - | 2 | 2 | 2 | 2 |

*—point A of Figure 8; **—point B of Figure 8; Or—orthoclase, Ab—albite, Csn—celsian, For—ferriorthoclase ($KFe^{3+}Si_3O_8$).

The compositions of alkali feldspar of coarse- and medium-grained rocks do not depend on whether or not a primary (not perthitic) albite is present in the rock. In any case, alkali feldspar is exsolved into pure albite and microcline, whose composition is $Or_{94-99}Ab_{1-6}$ (lines 3–7 in Figure 8a) with permanent impurities of Fe (up to 0.01 *apfu*) and Ba (up to 0.01 *apfu*).

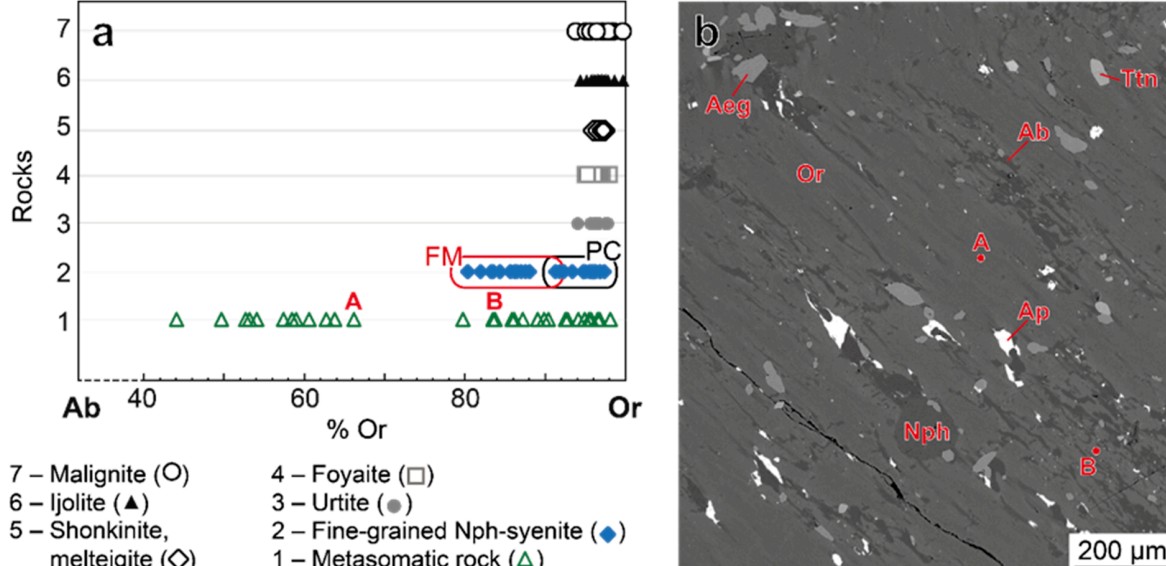

**Figure 8.** (**a**)—The composition of alkali feldspar from the rock of the Alluaiv site in the coordinates of albite (Ab)–orthoclase (Or). In fine-grained nepheline syenite: FM—fine-grained mass; PC—phenocrysts. In metasomatic rock: A, B–composition of alkali feldspar corresponding to points in Figure 8b. (**b**)—fragment of alkali feldspar crystal in metasomatic rock. BSE-image (156/98).

In fine-grained nepheline syenite, phenocrysts of alkali feldspar are also exsolved into pure albite and microcline $Or_{91-98}Ab_{2-9}$ (line 2 in Figure 8a). In the fine-grained mass, the composition of perthite-free microcline is $Or_{80-92}Ab_{6-17}$, with the same Ba amount (up to 0.01 *apfu*) and a higher Fe content (up to 0.03 *apfu*).

In metasomatic rocks, feldspar is exsolved into orthoclase and albite unevenly. There occur nearby grains of homogeneous feldspar $Or_{44-58}Ab_{55-41}$ and grains with small amounts of perthite (3–15 vol. %). The composition of such grains is $Or_{58-66}Ab_{42-34} + Ab_{100}$. An increase in the perthite quantity (up to 38 vol. %) leads to a decrease in Na content in the matrix (up to $Or_{98}Ab_2$). Sometimes, within the same grain, there are separated areas of homogeneous and perthite-bearing potassium feldspar (Figure 8a,b).

Albite lamellae in alkali feldspar from all types of rocks have the same composition: $Ab_{100}$. The composition of primary albite ranges as $Ab_{94-99}Or_{1-6}$ and does not depend on the type of rock. An impurity of $Fe_2O_3$, which reaches maximum values (up to 0.30 wt. %) in albite from fine-grained nepheline syenite, is characteristic.

### 4.3.2. Nepheline

Representative analyses of nepheline are shown in Table 2, while all available compositions are shown in Figure 9 in the coordinates of Nph (nepheline)–Ks (calsilite)–Qtz ($SiO_2$) and in Supplementary Materials (Table S5). Morphological types of nepheline are clearly separated by the content of Qtz endmember. The composition of nepheline-I is $Nph_{67-78}Ks_{15-23}Qtz_{1-12}$, while nepheline-II contains more Qtz-component: $Nph_{61-74}Ks_{13-21}Qtz_{9-23}$. In these intervals, the compositions of nepheline from most rocks of the Eudialyte Complex are evenly distributed, except for poikilitic rocks and fine-grained nepheline syenite, where nepheline is enriched with $SiO_2$. In fine-grained nepheline syenite, the compositions of nepheline from fine-grained mass and phenocrysts do not differ.

**Table 2.** Representative microprobe analyses of nepheline, wt. %.

| Rock | Malignite | | Ijolite | | Shonkinite, Melteigite | | Foyaite | | Urtite | | Fine-Grained Nph-syenite | | Metasomatic Rock | |
|---|---|---|---|---|---|---|---|---|---|---|---|---|---|---|
| Drill hole | 117 | 154 | 153 | 44 | 117 | 153 | 117 | 157 | 230 | 117 | 147 | 160 | 156 | 156 |
| Deep, m | 121 | 85 | 178 | 108 | 111 | 36 | 217 | 164 | 239 | 208 | 1 | 148 | 137 | 157 |
| $SiO_2$ | 44.76 | 41.79 | 42.71 | 45.72 | 43.14 | 44.58 | 42.38 | 44.74 | 43.12 | 41.45 | 42.20 | 43.61 | 44.35 | 45.81 |
| $Al_2O_3$ | 30.65 | 32.38 | 32.73 | 31.08 | 31.15 | 30.78 | 32.79 | 29.88 | 33.38 | 32.78 | 33.62 | 29.40 | 31.69 | 30.78 |
| $Fe_2O_3$ | 1.44 | 0.16 | 0.18 | 1.63 | 1.53 | 1.25 | 0.16 | 1.87 | 0.26 | 0.15 | 0.12 | 1.91 | 1.77 | 1.09 |
| CaO | b.d. | b.d. | b.d. | b.d. | b.d. | b.d. | b.d. | b.d. | b.d. | b.d. | b.d. | b.d. | b.d. | 0.03 |
| $K_2O$ | 5.44 | 6.63 | 6.71 | 6.08 | 5.85 | 5.81 | 6.51 | 5.42 | 6.62 | 6.67 | 7.25 | 5.36 | 5.47 | 5.35 |
| $Na_2O$ | 16.23 | 16.23 | 16.36 | 15.95 | 16.54 | 16.16 | 15.23 | 16.13 | 15.56 | 15.27 | 15.14 | 15.69 | 15.91 | 15.17 |
| Sum | 98.52 | 97.19 | 98.68 | 100.45 | 98.21 | 98.58 | 97.08 | 98.04 | 98.93 | 96.31 | 98.32 | 95.97 | 99.20 | 98.24 |
| Formula based on 8 oxygen atoms, *apfu* | | | | | | | | | | | | | | |
| Si | 4.35 | 4.15 | 4.18 | 4.37 | 4.24 | 4.34 | 4.19 | 4.38 | 4.18 | 4.14 | 4.14 | 4.36 | 4.28 | 4.43 |
| Al | 3.51 | 3.79 | 3.77 | 3.50 | 3.61 | 3.53 | 3.82 | 3.45 | 3.82 | 3.86 | 3.88 | 3.47 | 3.61 | 3.51 |
| $Fe^{3+}$ | 0.11 | 0.01 | 0.01 | 0.12 | 0.11 | 0.09 | 0.01 | 0.14 | 0.02 | 0.01 | 0.01 | 0.14 | 0.13 | 0.08 |
| K | 0.67 | 0.84 | 0.84 | 0.74 | 0.73 | 0.72 | 0.82 | 0.68 | 0.82 | 0.85 | 0.91 | 0.68 | 0.67 | 0.66 |
| Na | 3.06 | 3.13 | 3.10 | 2.95 | 3.15 | 3.05 | 2.92 | 3.06 | 2.93 | 2.96 | 2.88 | 3.04 | 2.98 | 2.85 |
| Sum | 11.71 | 11.93 | 11.90 | 11.67 | 11.84 | 11.73 | 11.76 | 11.70 | 11.77 | 11.82 | 11.81 | 11.70 | 11.67 | 11.53 |
| Mol. % endmembers | | | | | | | | | | | | | | |
| Nph | 70 | 75 | 74 | 68 | 74 | 70 | 70 | 70 | 70 | 71 | 69 | 70 | 69 | 64 |
| Ks | 16 | 20 | 20 | 17 | 17 | 17 | 19 | 15 | 20 | 21 | 22 | 16 | 16 | 15 |
| Qtz | 14 | 5 | 6 | 15 | 9 | 13 | 11 | 15 | 10 | 8 | 9 | 14 | 15 | 21 |

Nph—nepheline; Ks—kalsilite; Qtz—quartz.

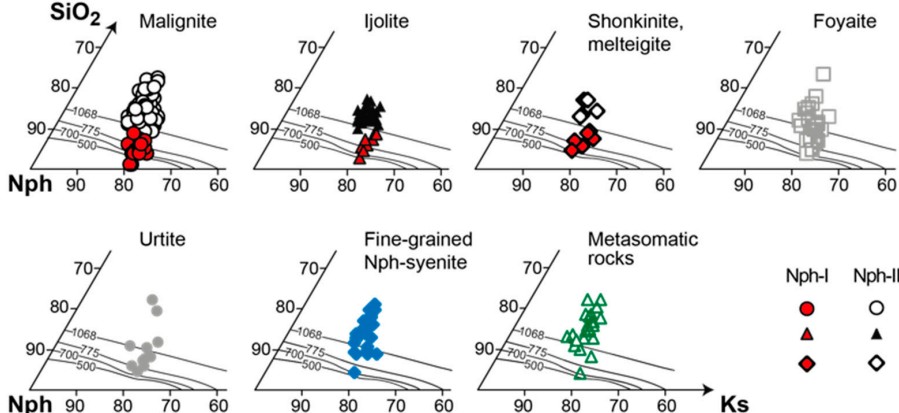

**Figure 9.** Nepheline compositions in the nepheline (Nph)–kalsilite (Ks)–quartz ($SiO_2$) triangle (on a mol. % basis) for the rocks of the Alluaiv site. The isotherms are from [31].

$Fe^{3+}$ (up to 0.16 *apfu*) is a permanent impurity in nepheline composition, and rare impurities, found in only 1% of the samples, are Ca (up to 0.01 *apfu*) and Ba (up to 0.004 *apfu*). According to the results of factor analysis of data on the nepheline composition (Figure 10), the main isomorphism scheme in this mineral can be written as:

$$\square_B + (Si^{4+} + Fe^{3+})_T \rightleftarrows K^+_B + 2Al^{3+}_T,$$

(if the composition of nepheline is expressed by the formula $A_4B_4T_8O_{16}$).

The same substitution is characteristic of nepheline of the Khibiny Massif [32,33]. Factor analysis has showed also that compositions corresponding to different types of rocks were divided into two almost isolated groups, in accordance with the main isomorphism scheme (Figure 10). Nepheline from fine-grained nepheline syenite and metasomatic rocks are enriched with iron. In malignite, ijolite, shonkinite, and melteigite, nepheline-I is rich in K and Al, while nepheline-II is enriched with iron. Most nepheline samples from foyaite and urtite are relatively rich in K and Al.

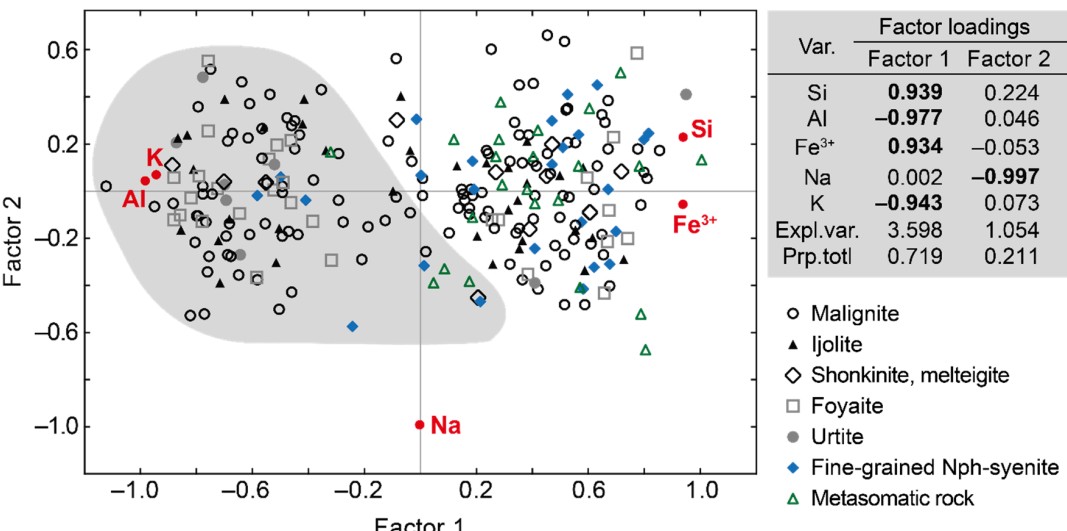

| Var. | Factor loadings | |
|---|---|---|
| | Factor 1 | Factor 2 |
| Si | **0.939** | 0.224 |
| Al | **−0.977** | 0.046 |
| $Fe^{3+}$ | **0.934** | −0.053 |
| Na | 0.002 | **−0.997** |
| K | **−0.943** | 0.073 |
| Expl.var. | 3.598 | 1.054 |
| Prp.totl | 0.719 | 0.211 |

○ Malignite
▲ Ijolite
◇ Shonkinite, melteigite
□ Foyaite
● Urtite
◆ Fine-grained Nph-syenite
△ Metasomatic rock

**Figure 10.** Results of factor analyses of data on the composition of nepheline from the Alliaiv site. Points on a gray background correspond to nepheline with inclusions of microcline and aegirine. Var.—variables; Expl.var.—Explained variance; Prp.totl—proportion of total variance. Factor loadings > |0.5| are shown in bold.

In meso- and melanocratic rocks (malignite, ijolite, shonkinite, and melteigite), contents of Si, $Fe^{3+}$, Al, K in nepheline-I correlate with $Fe^{3+}$ amount in coexisting microcline (Figure 11a). Interrelations between other components are weak because their ratio in the feldspar changed when perthites were formed. Also, there are no significant correlations between the compositions of nepheline-II and coexisting microcline. In foyaite and urtite, contents of Si, $Fe^{3+}$, Al, K in nepheline correlate with $Fe^{3+}$ amount in coexisting microcline. The compositions of coexisting nepheline and microcline from fine-grained nepheline syenite are interrelated, with positive correlations between the same elements (Figure 11b).

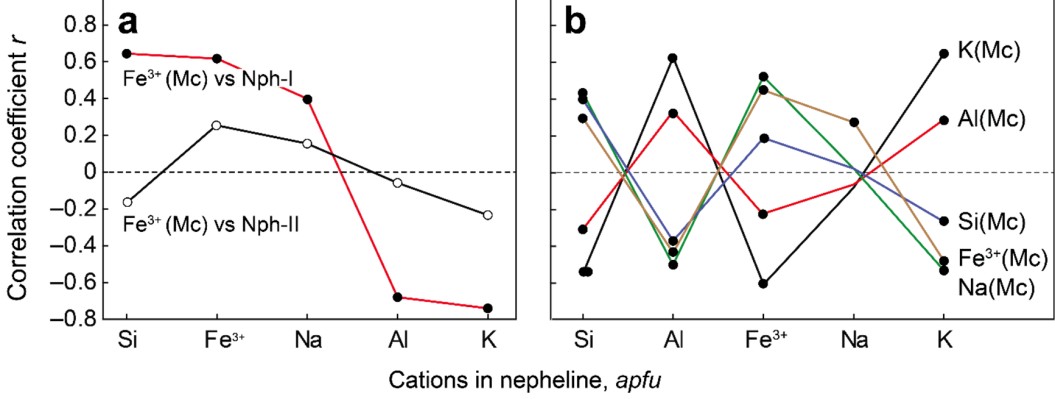

**Figure 11.** Correlation coefficients (r): (**a**)—between the content of $Fe^{3+}$ in microcline and cations in coexisting nepheline-I and nepheline-II; (**b**)—between cations in microcline (color lines) and cations in coexisting nepheline from fine-grained nepheline syenite. Black and empty dots indicate significant ($p \leq 0.05$) and insignificant correlation coefficients, respectively.

### 4.3.3. Clinopyroxenes

Representative chemical analyses of clinopyroxenes from the rocks of the Eudialyte Complex are presented in Table 3, and all analyses are in Table S6 (Supplementary Materials) and generated in Figure 12. In all types of rock, except for metasomatic varieties, clinopyroxenes are presented by

aegirine and aegirine-augite with high contents of Ti (up to 0.16 *apfu*), Zr (up to 0.05 *apfu*) and Al (up to 0.05 *apfu*).

**Table 3.** Representative microprobe analyses of clinopyroxenes, wt. %.

| Rock | Malignite | | Ijolite | | Shonkinite, Melteigite | | Foyaite | | Urtite | | Fine-Grained Nph-Syenite | | Metasomatic Rock | | | |
|---|---|---|---|---|---|---|---|---|---|---|---|---|---|---|---|---|
| Drill hole | 117 | 228 | 33 | 160 | 153 | 222 | 154 | 157 | 117 | 156 | 154 | 157 | 156 | 156 | 156 | 156 |
| Deep, m | 121 | 84 | 211 | 120 | 8 | 177 | 35 | 155 | 235 | 56 | 226 | 36 | 89 A** | 89 B | 89 C | 89 D |
| $SiO_2$ | 52.12 | 52.66 | 50.68 | 52.54 | 49.96 | 53.23 | 51.75 | 52.36 | 51.97 | 51.49 | 51.34 | 52.12 | 51.91 | 50.61 | 51.46 | 51.78 |
| $ZrO_2$ | 0.74 | 0.75 | 0.71 | 0.61 | 0.53 | 0.41 | 0.78 | 0.70 | 0.76 | 0.61 | 0.98 | 0.64 | b.d. | 0.27 | 1.05 | 0.90 |
| $TiO_2$ | 3.06 | 2.97 | 3.06 | 2.54 | 2.74 | 3.84 | 2.70 | 2.35 | 2.77 | 4.26 | 2.97 | 2.74 | 0.97 | 1.32 | 3.17 | 4.01 |
| $Al_2O_3$ | 0.79 | 0.89 | 0.71 | 0.80 | 0.87 | 0.80 | 0.77 | 0.97 | 0.86 | 0.89 | 0.76 | 0.78 | 1.33 | 1.12 | 0.90 | 0.87 |
| $V_2O_3$ | 0.05 | b.d. | b.d. | b.d. | b.d. | b.d. | b.d. | b.d. | b.d. | b.d. | b.d. | b.d. | b.d. | b.d. | b.d. | b.d. |
| CaO | 2.67 | 2.69 | 2.74 | 3.04 | 1.68 | 1.59 | 2.57 | 3.78 | 1.90 | 2.49 | 3.17 | 2.71 | 17.58 | 19.96 | 5.71 | 3.46 |
| MgO | 1.54 | 1.90 | 1.79 | 2.11 | 1.12 | 1.22 | 1.64 | 2.17 | 1.46 | 1.47 | 1.72 | 1.70 | 12.59 | 10.77 | 3.03 | 2.12 |
| FeO | 23.60 | 23.37 | 23.61 | 23.13 | 24.91 | 24.87 | 24.13 | 24.13 | 23.70 | 24.43 | 23.49 | 24.76 | 8.32 | 10.08 | 20.85 | 22.53 |
| MnO | 0.40 | 0.57 | 0.45 | 0.53 | 0.40 | 0.48 | 0.48 | 0.44 | 0.33 | 0.46 | 0.45 | 0.41 | 0.66 | 0.65 | 0.47 | 0.62 |
| ZnO | b.d. | b.d. | b.d. | b.d. | b.d. | b.d. | b.d. | b.d. | 0.08 | b.d. | b.d. | b.d. | b.d. | b.d. | b.d. | b.d. |
| $Na_2O$ | 12.28 | 12.57 | 12.05 | 12.44 | 13.41 | 13.40 | 12.60 | 12.20 | 13.15 | 12.16 | 12.59 | 12.79 | 1.85 | 2.91 | 10.48 | 12.04 |
| $K_2O$ | b.d. | b.d. | b.d. | b.d. | b.d. | b.d. | 0.10 | b.d. | b.d. | b.d. | b.d. | b.d. | b.d. | b.d. | b.d. | b.d. |
| Sum | 97.25 | 98.37 | 95.81 | 97.72 | 95.61 | 99.83 | 97.50 | 99.08 | 96.97 | 98.27 | 97.47 | 98.62 | 95.21 | 97.69 | 97.12 | 98.32 |
| Formula based on 4 cations and 6 oxygen atoms, *apfu* | | | | | | | | | | | | | | | | |
| Si | 1.99 | 1.98 | 1.96 | 1.98 | 1.92 | 1.97 | 1.96 | 1.96 | 1.97 | 1.96 | 1.95 | 1.95 | 2.02 | 1.93 | 1.98 | 1.96 |
| Zr | 0.01 | 0.01 | 0.01 | 0.01 | 0.01 | 0.01 | 0.01 | 0.01 | 0.01 | 0.01 | 0.02 | 0.01 | – | 0.01 | 0.02 | 0.02 |
| Ti | 0.09 | 0.08 | 0.09 | 0.07 | 0.08 | 0.11 | 0.08 | 0.07 | 0.08 | 0.12 | 0.09 | 0.08 | 0.03 | 0.04 | 0.09 | 0.11 |
| Al | 0.04 | 0.04 | 0.03 | 0.04 | 0.04 | 0.04 | 0.03 | 0.04 | 0.04 | 0.04 | 0.03 | 0.03 | 0.06 | 0.05 | 0.04 | 0.04 |
| Ca | 0.11 | 0.11 | 0.11 | 0.12 | 0.07 | 0.06 | 0.11 | 0.15 | 0.08 | 0.10 | 0.13 | 0.11 | 0.73 | 0.81 | 0.24 | 0.14 |
| Mg | 0.09 | 0.11 | 0.10 | 0.12 | 0.06 | 0.07 | 0.09 | 0.12 | 0.08 | 0.08 | 0.10 | 0.10 | 0.73 | 0.61 | 0.17 | 0.12 |
| $Fe^{2+}$ | 0.06 | 0.01 | 0.02 | - | - | 0.02 | - | - | - | 0.10 | - | - | 0.27 | 0.09 | 0.10 | 0.04 |
| $Fe^{3+}$ | 0.69 | 0.72 | 0.74 | 0.73 | 0.80 | 0.76 | 0.77 | 0.75 | 0.75 | 0.68 | 0.75 | 0.78 | - | 0.23 | 0.57 | 0.67 |
| Mn | 0.01 | 0.02 | 0.02 | 0.02 | 0.01 | 0.02 | 0.02 | 0.01 | 0.01 | 0.02 | 0.02 | 0.01 | 0.02 | 0.02 | 0.02 | 0.02 |
| Na | 0.91 | 0.92 | 0.91 | 0.91 | 1.00 | 0.96 | 0.93 | 0.88 | 0.97 | 0.90 | 0.93 | 0.93 | 0.14 | 0.22 | 0.78 | 0.88 |
| K | - | - | - | - | - | - | 0.01 | - | - | - | - | - | - | - | - | - |
| Mol % endmembers | | | | | | | | | | | | | | | | |
| Q | 13 | 11 | 12 | 12 | 6 | 7 | 10 | 13 | 8 | 14 | 11 | 10 | 100 | 78 | 25 | 15 |
| Aeg | 83 | 84 | 84 | 84 | 89 | 89 | 86 | 82 | 88 | 81 | 85 | 86 | - | 18 | 70 | 81 |
| Jd | 4 | 5 | 4 | 4 | 5 | 4 | 4 | 5 | 4 | 5 | 4 | 4 | - | 4 | 5 | 4 |

*—pyroxene compositions at points A–D correspond to Figure 12b; **—endmember composition 89A: $Wo_{42}En_{41}Fs_{17}$.

The factor analysis of the clinopyroxene compositions (Figure 12a) revealed two main schemes of isomorphism:

$$Na^+ + Ti^{4+} + Al^{3+} \rightleftarrows Ca^{2+} + Mg^{2+} + Zr^{4+};$$

$$Fe^{3+} + (Al, Fe)^{3+} \rightleftarrows Fe^{2+} + Si^{4+}.$$

The zirconium content goes up with an increase in Ca and Mg and reaches maximum values in aegirine-augite. High concentrations of titanium, on the contrary, are characteristic of aegirine. The chemical composition of clinopyroxenes in rocks of the Eudialyte Complex (except for metasomatic rocks) varies widely, but does not depend on the type of rock (Figure 12a,b).

It is important to note the constant admixture of manganese (up to 0.04 *apfu*) in clinopyroxenes (see Table 3). Potassium impurity (up to 0.01 *apfu*) was recorded in 7% of studied samples, while impurities of vanadium (up to 0.002 *apfu*) and zinc (up to 0.02 *apfu*) occur in 2% of the samples.

In metasomatic rocks, resorbed augite relics (point A in Figure 13b) are surrounded at first by aegirine-augite (points B and C), and then by aegirine (point D). Factor analysis of the data on the composition of clinopyroxenes from these rocks (Figure 13a) managed to separate relics of (aegirine)-augite from later aegirine enriched with titanium and zirconium.

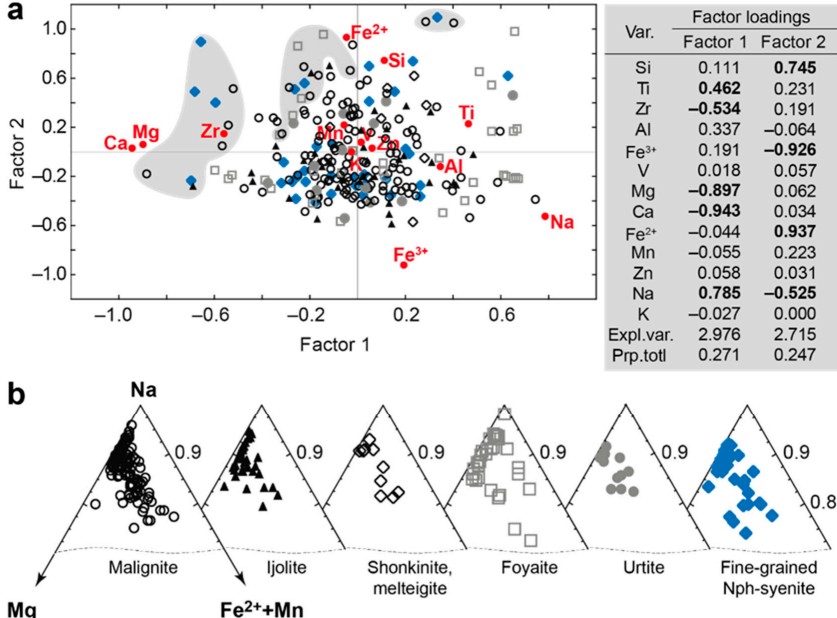

**Figure 12.** Chemical composition of clinopyroxenes from magmatic rocks of the Alluaiv site: (**a**)—results of factor analyses of data on the composition of clinopyroxenes from the Alluaiv site. Points on a gray background correspond to aegirine-augite, the other points are of aegirine; (**b**)—compositional variation of clinopyroxene in triangular system Mg–($Fe^{2+}$ + Mn)–Na. Var.—variables; Expl.var.—Explained variance; Prp.totl—proportion of total variance. Factor loadings > |0.5| are shown in bold.

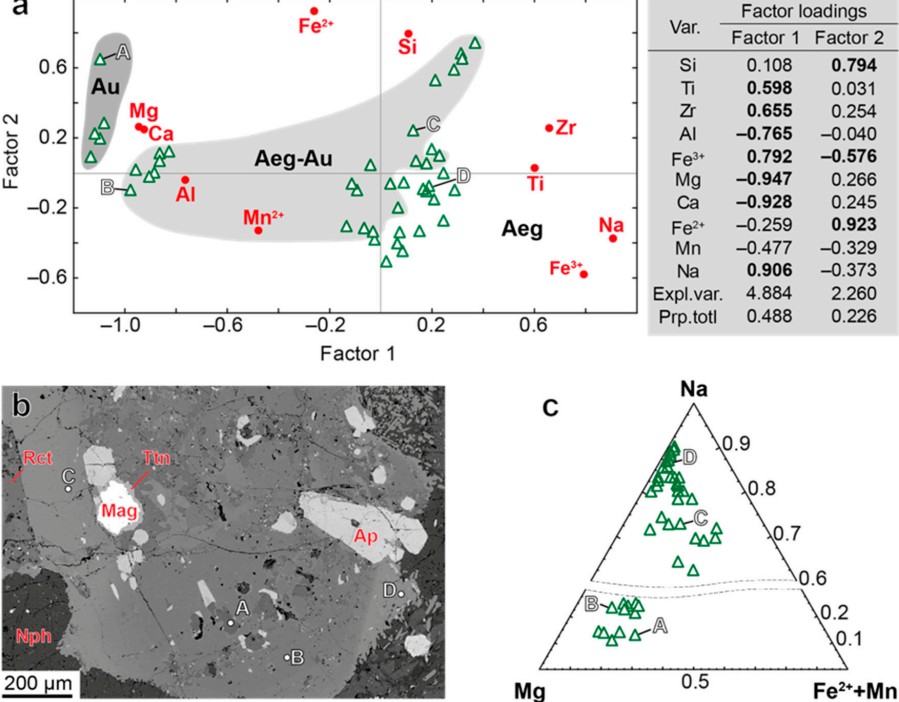

**Figure 13.** Chemical composition of clinopyroxenes from metasomatic rocks of the Alluaiv site: (**a**)—results of factor analyses. Points on a gray background correspond to augite (Au) and aegirine-augite (Aeg-Au), with other points corresponding to aegirine (Aeg). Points A–D correspond to Figure 13b,c. Var.—variables; Expl.var.—Explained variance; Prp.totl—proportion of total variance. Factor loadings > |0.5| are shown in bold. (**b**)—relicts of augite surrounded by rims of aegirine-augite and aegirine in a metasomatic rock 156/89. The compositions corresponding to points A–D are shown in Figure 13a,c; (**c**)—compositional variation of clinopyroxenes in Mg–($Fe^{2+}$ + Mn)–Na system.

There are significant correlations between a sodium content in nepheline (morphological types I and II) and contents of sodium ($r$ = 0.350), ferric iron ($r$ = 0.354), ferrous iron ($r$ = −0.460) and silicon ($r$ = −0.456) in clinopyroxenes in meso- and melanocratic rocks. Similar correlations have been established for nepheline and clinopyroxenes from fine-grained nepheline syenite.

### 4.3.4. Amphiboles

Representative analyses of amphiboles are given in Table 4, and all analyses are in Table S7 (Supplementary Materials). Formula calculations for amphiboles were made on the base of 16 cations and 23 oxygens. This calculation resulted in systematically slightly under-occupied C-sites and over-occupied A-sites, which may be explained by significant Li contents [34] that cannot be measured by an electron microprobe. A method of lithium content calculation proposed in [34] cannot be used, since the discrepancy between measured and calculated $Li_2O$ is quite significant at relatively high lithium content typical for amphiboles of the Lovozero Massif (up to 0.6 wt. % $Li_2O$ [24]). The cation excess in *C*-site and deficit in *A*-site were not found only in 37% (80 samples) of investigated amphiboles. The most of these amphiboles (70 samples) are magnesioarfvedsonite, the rest belong to arfvedsonite.

**Table 4.** Representative microprobe analyses of amphiboles, wt. %.

| Rock | Malignite | | Ijolite | | Shonkinite, Melteigite | | Foyaite | | Urtite | | Fine-Grained Nph-Syenite | | Metasomatic Rock | |
|---|---|---|---|---|---|---|---|---|---|---|---|---|---|---|
| Drill hole | 117 | 147 | 231 | 117 | 153 | 222 | 147 | 154 | 147 | 156 | 154 | 154 | 156 | 231 |
| Deep, m | 4 | 194 | 176 | 93 | 36 | 231 | 275 | 172 | 85 | 56 | 245 * | 245 ** | 89 *** | 146 |
| $SiO_2$ | 51.81 | 53.78 | 53.10 | 53.32 | 52.33 | 52.50 | 52.30 | 53.82 | 52.90 | 52.44 | 53.12 | 51.10 | 52.56 | 54.65 |
| $TiO_2$ | 1.76 | 1.50 | 1.36 | 1.54 | 1.63 | 1.77 | 1.56 | 1.02 | 1.47 | 1.31 | 1.64 | 0.86 | 0.56 | 2.01 |
| $Al_2O_3$ | 1.49 | 0.77 | 0.78 | 1.25 | 1.33 | 1.29 | 1.14 | 1.09 | 1.22 | 1.00 | 0.62 | 0.58 | 1.99 | 1.61 |
| FeO | 17.54 | 17.19 | 17.56 | 17.00 | 18.16 | 16.79 | 17.75 | 12.62 | 16.78 | 18.00 | 17.42 | 26.17 | 10.95 | 14.24 |
| MnO | 1.39 | 1.50 | 1.49 | 1.20 | 1.52 | 1.51 | 1.49 | 1.68 | 1.37 | 1.38 | 1.66 | 3.13 | 1.29 | 1.61 |
| MgO | 9.36 | 10.45 | 10.03 | 10.37 | 9.75 | 10.29 | 10.52 | 13.08 | 11.62 | 9.61 | 9.32 | 2.64 | 14.00 | 12.98 |
| CaO | 0.94 | 0.83 | 0.59 | 1.15 | 1.02 | 1.13 | 1.20 | 0.91 | 1.35 | 0.71 | 0.48 | 0.29 | 5.66 | 1.65 |
| ZnO | 0.07 | 0.05 | 0.06 | b.d. | 0.08 | 0.05 | b.d. | 0.05 | b.d. | b.d. | 0.08 | 0.07 | 0.07 | b.d. |
| $Na_2O$ | 9.30 | 9.50 | 9.17 | 9.26 | 8.90 | 9.37 | 9.26 | 9.41 | 8.92 | 8.78 | 10.10 | 8.46 | 6.87 | 9.12 |
| $K_2O$ | 1.65 | 1.55 | 1.56 | 1.68 | 1.76 | 1.67 | 1.61 | 1.53 | 1.64 | 1.67 | 1.68 | 3.11 | 1.40 | 1.57 |
| F | b.d. | b.d. | 2.10 | b.d. | b.d. | 1.50 | b.d. | b.d. | b.d. | b.d. | b.d. | b.d. | b.d. | 1.70 |
| O=F | 0.00 | 0.00 | −0.88 | 0.00 | 0.00 | −0.63 | 0.00 | 0.00 | 0.00 | 0.00 | 0.00 | 0.00 | 0.00 | −0.72 |
| Sum | 95.31 | 97.12 | 96.92 | 96.77 | 96.48 | 97.24 | 96.83 | 95.21 | 97.27 | 94.90 | 96.11 | 96.41 | 95.35 | 100.42 |
| Formula based on O + OH + F = 24 *apfu* and OH = 2 − 2Ti | | | | | | | | | | | | | | |
| Si (*T*) | 7.83 | 7.94 | 7.99 | 7.90 | 7.84 | 7.81 | 7.76 | 7.93 | 7.78 | 7.99 | 7.93 | 8.04 | 7.84 | 7.80 |
| Al (*T*) | 0.17 | 0.05 | 0.01 | 0.10 | 0.16 | 0.19 | 0.20 | 0.07 | 0.21 | 0.01 | 0.07 | - | 0.16 | 0.20 |
| Ti (*T*) | - | - | - | - | - | - | 0.04 | - | 0.01 | - | - | - | - | - |
| **Sum *T*** | **8.00** | **7.99** | **8.00** | **8.00** | **8.00** | **8.00** | **8.00** | **8.00** | **8.00** | **8.00** | **8.00** | **8.04** | **8.00** | **8.00** |
| Ti (*C*) | 0.20 | 0.17 | 0.15 | 0.17 | 0.18 | 0.20 | 0.13 | 0.11 | 0.15 | 0.15 | 0.18 | 0.10 | 0.06 | 0.22 |
| Al (*C*) | 0.09 | 0.08 | 0.12 | 0.12 | 0.08 | 0.04 | - | 0.12 | - | 0.16 | 0.03 | 0.11 | 0.19 | 0.07 |
| $Fe^{3+}$ (*C*) | 0.72 | 0.66 | 0.56 | 0.62 | 0.64 | 0.78 | 0.91 | 0.70 | 0.75 | 0.47 | 0.92 | 0.81 | - | 0.51 |
| Mn (*C*) | 0.18 | 0.19 | 0.19 | 0.15 | 0.19 | 0.19 | 0.19 | 0.21 | 0.17 | 0.18 | 0.21 | 0.42 | 0.16 | 0.19 |
| $Fe^{2+}$ (*C*) | 1.49 | 1.46 | 1.65 | 1.49 | 1.64 | 1.31 | 1.29 | 0.86 | 1.31 | 1.83 | 1.26 | 2.64 | 1.36 | 1.19 |
| Mg (*C*) | 2.11 | 2.30 | 2.25 | 2.29 | 2.18 | 2.28 | 2.33 | 2.87 | 2.55 | 2.18 | 2.07 | 0.62 | 3.11 | 2.76 |
| Zn (*C*) | 0.01 | - | 0.01 | - | 0.01 | - | - | 0.01 | - | - | 0.01 | 0.01 | 0.01 | - |
| **Sum *C*** | **4.80** | **4.86** | **4.93** | **4.84** | **4.92** | **4.80** | **4.85** | **4.88** | **4.93** | **4.97** | **4.68** | **4.71** | **4.89** | **4.94** |
| Ca (*B*) | 0.15 | 0.13 | 0.09 | 0.18 | 0.16 | 0.18 | 0.19 | 0.14 | 0.21 | 0.12 | 0.08 | 0.05 | 0.90 | 0.25 |
| Na (*B*) | 1.85 | 1.87 | 1.91 | 1.82 | 1.84 | 1.82 | 1.81 | 1.86 | 1.79 | 1.88 | 1.92 | 1.95 | 1.10 | 1.75 |
| **Sum *B*** | **2.00** | **2.00** | **2.00** | **2.00** | **2.00** | **2.00** | **2.00** | **2.00** | **2.00** | **2.00** | **2.00** | **2.00** | **2.00** | **2.00** |
| Na (*A*) | 0.88 | 0.85 | 0.77 | 0.84 | 0.74 | 0.88 | 0.85 | 0.83 | 0.76 | 0.71 | 1.00 | 0.63 | 0.89 | 0.77 |
| K (*A*) | 0.32 | 0.30 | 0.30 | 0.32 | 0.34 | 0.32 | 0.30 | 0.29 | 0.31 | 0.32 | 0.32 | 0.62 | 0.27 | 0.29 |
| **Sum *A*** | **1.20** | **1.15** | **1.07** | **1.16** | **1.08** | **1.20** | **1.15** | **1.12** | **1.07** | **1.03** | **1.30** | **1.25** | **1.16** | **1.06** |
| OH (*W*) | 2.00 | 2.00 | 1.00 | 2.00 | 2.00 | 1.29 | 2.00 | 2.00 | 2.00 | 2.00 | 2.00 | 2.00 | 2.00 | 1.23 |
| F (*W*) | - | - | 1.00 | - | - | 0.71 | - | - | - | - | - | - | - | 0.77 |
| **Sum *W*** | **2.00** | **2.00** | **2.00** | **2.00** | **2.00** | **2.00** | **2.00** | **2.00** | **2.00** | **2.00** | **2.00** | **2.00** | **2.00** | **2.00** |

*—core of zonal grain; **—edge of zonal grain; ***—rim around augite relict (see Figure 13b). *T,C,B,A,W*—sites in general formula $AB_2C_5T_8O_{22}W_2$ [27].

The lithium content determines the value of the ratio $Fe^{3+}/Fe^{2+}$ in amphiboles, because of following replacement: $Li_{M3} + Fe^{3+} \rightleftarrows Fe^{2+}{}_{M3} + Fe^{2+}$ [35]. In this study, measurements of Li content in amphiboles were not carried out; therefore, we do not discuss the $Fe^{3+}/Fe^{2+}$ ratio in amphiboles. As an alternative, we considered concentrations of major and minor elements in binary diagrams (Figure 14). Amphiboles from meso- and melanocratic rocks are characterized by an increased content of Ca (up to 0.28 *apfu*), Al (up to 0.30 *apfu*) and Ti (up to 0.28 *apfu*), while amphiboles from leucocratic rocks and fine-grained nepheline syenite are enriched in K(up to 0.71 *apfu*), Si and Mn (up to 0.52 *apfu*).

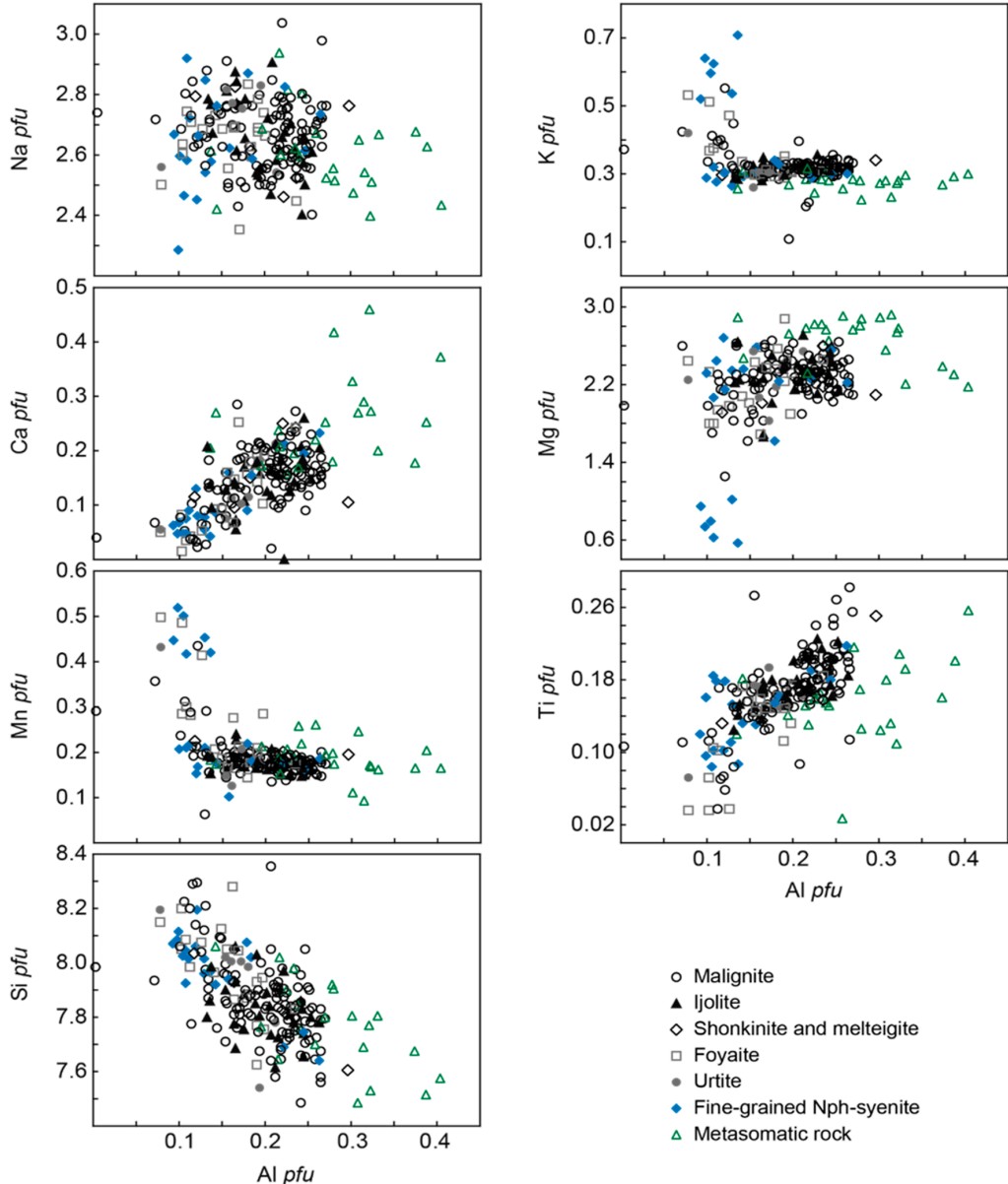

**Figure 14.** Binary diagrams showing major and minor element concentrations in amphiboles as a function of Al content.

In the metasomatic rocks, sodium-calcium (ferri-katophorite, richterite) and sodium (eckermannite) amphiboles along with aegirine-augite surround the augite relics. These amphiboles stand out clearly in binary diagrams (Figure 14) with high Ca (up to 0.46 *apfu*) and Al (up to 0.41 *apfu*) contents. Magnesioarfvedsonite is also characteristic of these rocks, but it forms symplectic intergrowths with nepheline (see Figure 5c,d).

In meso- and melanocratic rocks, cations in the composition of amphiboles (only samples without under-occupied C-sites and over-occupied A-sites) and coexisting clinopyroxenes correlate closely with each other, with positive correlations between the same elements (Figure 15).

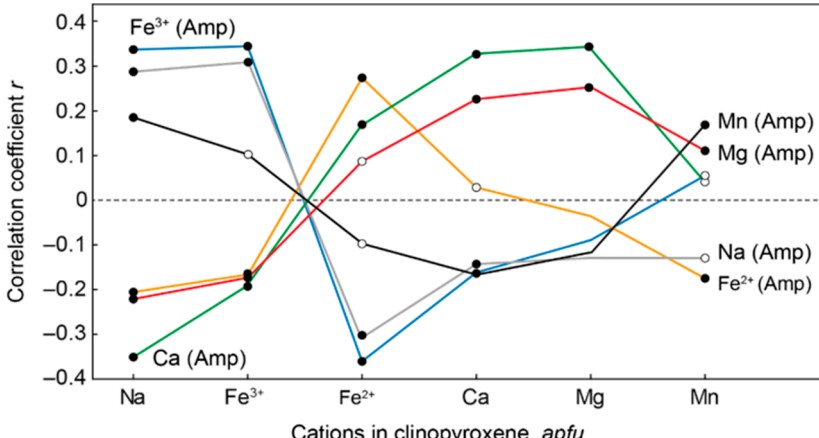

**Figure 15.** Correlation coefficients between the cations of coexisting amphiboles and clinopyroxenes. Black and empty dots indicate significant ($p \leq 0.05$) and insignificant correlation coefficients, respectively.

4.3.5. Eudialyte-Group Minerals (EGM)

Representative analyses of EGM and site occupancies calculated using the method of Johnsen and Grice [36] are given in Table 5, Table 6 and Table S8 (Suplementary Materials) according the IMA-accepted general formula for the eudialyte-group minerals [37,38]: $N_{15}[M(1)]_6[M(2)]_3[M(3)][M(4)]Z_3 (Si_{24}O_{72})O'_4X_2$, with $N$ = Na, Ca, K, Sr, REE, Ba, Mn, $H_3O^+$; $M(1)$ = Ca, Mn, REE, Na, Sr, Fe; $M(2)$ = Fe, Mn, Na, Zr, Ta, Ti, K, Ba, $H_3O^+$; $M(3, 4)$ = Si, Nb, Ti, W, Na; $Z$ = Zr, Ti, Nb; $O'$ = O, $OH^-$, $H_2O$; $X$ = $H_2O$, $Cl^-$, $F^-$, $OH^-$, $CO_3^{2-}$, $SO_4^{2-}$, $SiO_4^{4-}$.

The high zirconium content (above 3 *apfu*) is the main feature of the most EGM samples. As a result, there is an excess of zirconium remains after $Z$ position filling. There remains a possibility of zirconium to enter the $M(2)$-site, which is not completely filled in the majority of the samples (Figure 16) and in the $M(3)$-site, where there is also a constant deficit of cations. The results of the factor analysis of EGM compositions (Figure 17) indicate that Si is replaced by Zr; therefore, we assigned excess zirconium to the $M(3)$-site. Zirconium dominates in the $M(3)$-site in 22% of EGM samples, Si prevails in 75% samples, and niobium predominates in 3% samples. However, the EGM classification [37] does not includes EGM varieties with a predominant Zr in any position other than $Z$, and such EGM from the Lovozero Eudialyte Complex, of course, require an additional study. The remaining samples belong to the eudialyte solid solution (eudialyte$_{ss}$)–kentbrooksite series.

According to the factor analysis results (Figure 17), the main scheme of isomorphic substitutions in eudialyte is as follows:

$$Zr^{4+} + Al^{3+} + Ti^{4+} + Fe^{2+} + 3Na^+ \rightleftarrows Si^{4+} + Nb^{5+} + Mn^{2+} + REE^{3+} + (Ca, Sr)^{2+}.$$

With an increase in the kentbrooksite endmember content, the concentrations of rare-earth elements increase, and with an increase in eudialyte$_{ss}$ endmember fraction, the contents of Zr and Al increase. The anomalous Zr-rich EGM (Zr-rich eudialyte$_{ss}$) are likely to have resulted from a sharp shift of the main equilibrium to the left. In zonal EGM grains (see Figure 3e), transition from the grain cores to their marginal zones occurs on the following schemes: eudialyte$_{ss}$→ kentbrooksite or Zr-rich eudialyte$_{ss}$ → eudialyte$_{ss}$. Zonal EGM crystals were found only in meso-and melanocratic rocks; in other rocks, EGM form homogeneous grains.

There are no significant differences in the composition of EGM from meso-, melano-, and leucocratic rocks. On average, EGM from meso- and melanocratic rocks contain more $Fe^{2+}$ (median 1.18 *apfu*) and less Mn (median 1.04 *apfu*) than EGM from leucocratic rocks (median 0.65 Fe *pfu* and 1.16 Mn *pfu*). EGM from metasomatic rocks are enriched with Ca, Sr, Nb, and Fe.

In the meso- and melanocratic rocks, the compositions of coexisting clinopyroxenes and EGM correlate with each other. With an increase in magnesium and calcium in clinopyroxene (diopside endmember), the calcium content in EGM growths linearly (*r* = 0.444 and 0.441, respectively).

**Table 5.** Representative microprobe analyses of EGM, wt. %.

| Rock. | Malignite | | Ijolite | | Shonkinite | | Foyaite | | Urtite | | Fine-grained Nph- Syenite | | Metasomatic Rock | |
|---|---|---|---|---|---|---|---|---|---|---|---|---|---|---|
| Drill hole | 28 | 28 | 117 | 117 | 44 | 44 | 117 | 156 | 147 | 154 | 154 | 160 | 154 | 156 |
| Deep, m | 153 * | 153 ** | 47 * | 47 ** | 50 * | 50 ** | 215 | 216 | 85 | 144 | 226 | 148 | 197 | 118 |
| $Nb_2O_5$ | 0.49 | 0.39 | 0.67 | 0.73 | 0.72 | 0.71 | 0.35 | 1.22 | 0.81 | 0.71 | 0.36 | 1.11 | 0.64 | 1.62 |
| $SiO_2$ | 49.99 | 51.75 | 49.45 | 50.97 | 53.07 | 50.44 | 51.81 | 49.92 | 49.34 | 51.24 | 53.51 | 49.48 | 51.02 | 49.33 |
| $ZrO_2$ | 12.54 | 11.69 | 13.95 | 12.79 | 12.04 | 13.90 | 13.03 | 11.78 | 12.90 | 11.20 | 14.71 | 11.65 | 12.93 | 12.80 |
| $TiO_2$ | 0.66 | 0.46 | 0.61 | 0.36 | 0.38 | 0.66 | 0.65 | 0.52 | 0.70 | 0.75 | 1.01 | 0.56 | 0.65 | 0.29 |
| $Al_2O_3$ | 0.22 | 0.15 | 0.16 | 0.10 | 0.07 | 0.22 | 0.17 | 0.23 | 0.22 | 0.10 | 0.14 | 0.21 | 0.24 | 0.16 |
| $Y_2O_3$ | b.d. | b.d. | b.d. | b.d. | b.d. | b.d. | b.d. | b.d. | b.d. | b.d. | b.d. | b.d. | b.d. | 0.84 |
| $La_2O_3$ | 0.19 | 0.40 | 0.29 | 0.27 | 0.30 | 0.25 | 0.46 | 0.42 | 0.25 | 0.61 | 0.37 | 0.33 | b.d. | 0.25 |
| $Ce_2O_3$ | 0.53 | 0.85 | 0.57 | 0.69 | 0.58 | 0.74 | 0.98 | 0.94 | 0.62 | 1.73 | 1.24 | 0.94 | 0.51 | 0.56 |
| $Nd_2O_3$ | 0.32 | 0.33 | 0.37 | 0.39 | 0.17 | 0.19 | 0.33 | 0.39 | 0.19 | 0.68 | 0.39 | 0.32 | b.d. | 0.25 |
| CaO | 7.48 | 8.71 | 6.27 | 7.19 | 9.04 | 6.20 | 6.92 | 7.96 | 6.34 | 7.33 | 6.49 | 8.09 | 10.31 | 8.54 |
| MgO | 0.16 | b.d. | 0.08 | 0.03 | 0.09 | 0.12 | b.d. | b.d. | 0.07 | b.d. | b.d. | b.d. | b.d. | 0.20 |
| FeO | 3.42 | 1.87 | 2.90 | 3.00 | 3.11 | 3.73 | 1.07 | 5.00 | 3.03 | 0.32 | 2.66 | 4.54 | 1.64 | 4.73 |
| MnO | 2.29 | 2.89 | 1.95 | 2.10 | 2.31 | 1.93 | 2.96 | 1.95 | 2.75 | 4.06 | 2.64 | 2.18 | 1.58 | 2.44 |
| SrO | 1.30 | 1.28 | 2.05 | 1.97 | 1.84 | 1.55 | 3.13 | 2.17 | 1.48 | 2.20 | 2.28 | 2.32 | 1.28 | 3.70 |
| BaO | 0.18 | 0.22 | b.d. | b.d. | 0.15 | b.d. | 0.76 | b.d. | 0.26 | 0.62 | 0.20 | 0.17 | 0.19 | 0.14 |
| $Na_2O$ | 16.02 | 15.97 | 15.32 | 13.19 | 12.46 | 16.14 | 13.39 | 15.51 | 14.68 | 15.47 | 9.49 | 14.69 | 12.50 | 11.35 |
| $K_2O$ | 0.24 | 0.18 | 0.29 | 0.22 | 0.19 | 0.29 | 0.23 | 0.27 | 0.25 | 0.47 | 0.19 | 0.22 | 0.28 | 0.32 |
| Cl | 1.51 | 1.17 | 1.30 | 1.08 | 1.04 | 1.27 | 1.46 | 1.56 | 1.27 | 0.98 | 1.67 | 1.44 | 1.27 | 1.58 |
| O=Cl | −0.34 | −0.26 | −0.29 | −0.24 | −0.23 | −0.29 | −0.33 | −0.35 | −0.29 | −0.22 | −0.38 | −0.33 | −0.29 | −0.36 |
| Total | 95.71 | 96.86 | 94.63 | 93.75 | 96.27 | 96.76 | 95.91 | 97.93 | 93.58 | 97.26 | 95.28 | 96.47 | 93.48 | 97.15 |
| Formula based on Si + Zr + Ti + Nb + Al + Hf = 29 *apfu* | | | | | | | | | | | | | | |
| Nb | 0.11 | 0.09 | 0.15 | 0.16 | 0.16 | 0.16 | 0.08 | 0.28 | 0.19 | 0.16 | 0.08 | 0.26 | 0.14 | 0.37 |
| Si | 25.40 | 25.81 | 25.07 | 25.52 | 25.81 | 25.10 | 25.46 | 25.45 | 25.20 | 25.75 | 25.13 | 25.48 | 25.34 | 25.23 |
| Zr | 3.11 | 2.84 | 3.45 | 3.12 | 2.85 | 3.37 | 3.12 | 2.93 | 3.21 | 2.75 | 3.37 | 2.92 | 3.13 | 3.19 |
| Ti | 0.25 | 0.17 | 0.23 | 0.13 | 0.14 | 0.25 | 0.24 | 0.20 | 0.27 | 0.28 | 0.36 | 0.22 | 0.24 | 0.11 |
| Al | 0.13 | 0.09 | 0.09 | 0.06 | 0.04 | 0.13 | 0.10 | 0.14 | 0.13 | 0.06 | 0.08 | 0.13 | 0.14 | 0.10 |
| Y | - | - | - | - | - | - | - | - | - | - | - | - | - | 0.23 |
| La | 0.04 | 0.07 | 0.05 | 0.05 | 0.05 | 0.05 | 0.08 | 0.08 | 0.05 | 0.11 | 0.06 | 0.06 | 0.00 | 0.05 |
| Ce | 0.10 | 0.16 | 0.11 | 0.13 | 0.10 | 0.14 | 0.18 | 0.18 | 0.12 | 0.32 | 0.21 | 0.18 | 0.09 | 0.11 |
| Nd | 0.06 | 0.06 | 0.07 | 0.07 | 0.03 | 0.03 | 0.06 | 0.07 | 0.04 | 0.12 | 0.07 | 0.06 | 0.00 | 0.05 |
| Ca | 4.07 | 4.65 | 3.41 | 3.86 | 4.71 | 3.30 | 3.64 | 4.35 | 3.47 | 3.95 | 3.27 | 4.46 | 5.49 | 4.68 |
| Mg | 0.12 | 0.00 | 0.06 | 0.02 | 0.06 | 0.09 | - | - | 0.05 | - | - | - | - | 0.15 |
| $Fe^{2+}$ | 1.45 | 0.78 | 1.23 | 1.25 | 1.26 | 1.55 | 0.44 | 2.13 | 1.29 | 0.13 | 1.04 | 1.95 | 0.68 | 2.02 |
| Mn | 0.99 | 1.22 | 0.84 | 0.89 | 0.95 | 0.81 | 1.23 | 0.84 | 1.19 | 1.73 | 1.05 | 0.95 | 0.67 | 1.05 |
| Sr | 0.38 | 0.37 | 0.60 | 0.57 | 0.52 | 0.45 | 0.89 | 0.64 | 0.44 | 0.64 | 0.62 | 0.69 | 0.37 | 1.10 |
| Ba | 0.04 | 0.04 | - | - | 0.03 | - | 0.15 | - | 0.05 | 0.12 | 0.04 | 0.03 | 0.04 | 0.03 |
| Na | 15.77 | 15.44 | 15.06 | 12.81 | 11.75 | 15.57 | 12.76 | 15.33 | 14.54 | 15.08 | 8.64 | 14.67 | 12.03 | 11.25 |
| K | 0.16 | 0.12 | 0.19 | 0.14 | 0.12 | 0.18 | 0.14 | 0.18 | 0.16 | 0.30 | 0.12 | 0.15 | 0.18 | 0.21 |
| Cl | 1.30 | 0.99 | 1.12 | 0.91 | 0.86 | 1.07 | 1.22 | 1.35 | 1.10 | 0.83 | 1.33 | 1.25 | 1.07 | 1.37 |
| Sum | 52.18 | 51.91 | 50.61 | 48.78 | 48.58 | 51.17 | 48.58 | 52.79 | 50.40 | 51.50 | 44.11 | 52.21 | 48.55 | 49.92 |

*—core of zonal grain; **—edge of zonal grain.

**Table 6.** Site occupancies of EGM from Table 5, *apfu.*

| Rock | Malignite | | Ijolite | | Shonkinite | | Foyaite | | Urtite | | Fine-grained Nph-Syenite | | Metasomatic Rock | |
|---|---|---|---|---|---|---|---|---|---|---|---|---|---|---|
| Drill hole | 28 | 28 | 117 | 117 | 44 | 44 | 117 | 156 | 147 | 154 | 154 | 160 | 154 | 156 |
| Interval, m | 153 * | 153 ** | 47 * | 47 ** | 50 * | 50 ** | 215 | 216 | 85 | 144 | 226 | 148 | 197 | 118 |
| Al(*M*4) | 0.13 | 0.09 | 0.09 | 0.06 | 0.04 | 0.13 | 0.10 | 0.14 | 0.13 | 0.06 | 0.08 | 0.13 | 0.14 | 0.10 |
| Si (*M*4) | 0.87 | 0.91 | 0.91 | 0.94 | 0.96 | 0.87 | 0.90 | 0.86 | 0.87 | 0.94 | 0.92 | 0.87 | 0.86 | 0.90 |
| **Sum** M4 | **1.00** | **1.00** | **1.00** | **1.00** | **1.00** | **1.00** | **1.00** | **1.00** | **1.00** | **1.00** | **1.00** | **1.00** | **1.00** | **1.00** |
| Zr (*Z*) | 3.00 | 2.84 | 3.00 | 3.00 | 2.85 | 3.00 | 3.00 | 2.93 | 3.00 | 2.75 | 3.00 | 2.92 | 3.00 | 3.00 |
| Ti (*Z*) | - | 0.16 | - | - | 0.14 | - | - | 0.07 | - | 0.25 | - | 0.08 | - | - |
| Nb (*Z*) | - | - | - | - | 0.01 | - | - | - | - | - | - | - | - | - |
| Sum *Z* | 3.00 | 3.00 | 3.00 | 3.00 | 3.00 | 3.00 | 3.00 | 3.00 | 3.00 | 3.00 | 3.00 | 3.00 | 3.00 | 3.00 |
| Nb (*M*3) | 0.11 | 0.09 | 0.15 | 0.16 | 0.15 | 0.16 | 0.08 | 0.28 | 0.19 | 0.16 | 0.08 | 0.26 | 0.14 | 0.37 |
| Si (*M*3) | 0.53 | 0.90 | 0.17 | 0.58 | 0.85 | 0.22 | 0.56 | 0.59 | 0.33 | 0.81 | 0.20 | 0.60 | 0.48 | 0.32 |
| Ti(*M*3) | 0.25 | 0.01 | 0.23 | 0.13 | - | 0.25 | 0.24 | 0.13 | 0.27 | 0.03 | 0.36 | 0.14 | 0.24 | 0.11 |
| Zr (*M*3) | 0.11 | - | 0.45 | 0.12 | - | 0.37 | 0.12 | - | 0.21 | - | 0.37 | - | 0.13 | 0.19 |
| **Sum *M*3** | **1.00** | **1.00** | **1.00** | **1.00** | **1.00** | **1.00** | **1.00** | **1.00** | **1.00** | **1.00** | **1.00** | **1.00** | **1.00** | **1.00** |
| Fe (*M*2) | 1.45 | 0.78 | 1.23 | 1.25 | 1.26 | 1.55 | 0.44 | 2.13 | 1.29 | 0.13 | 1.04 | 1.95 | 0.68 | 2.02 |
| Mn (*M*2) | 0.99 | 1.22 | 0.84 | 0.89 | 0.95 | 0.81 | 1.23 | 0.84 | 1.19 | 1.73 | 1.05 | 0.95 | 0.67 | 0.82 |
| Mg (*M*2) | 0.12 | - | 0.06 | 0.02 | 0.06 | 0.09 | - | - | 0.05 | - | - | - | - | 0.15 |
| **Sum** M2 | **2.56** | **2.00** | **2.13** | **2.16** | **2.28** | **2.45** | **1.67** | **2.97** | **2.54** | **1.86** | **2.10** | **2.90** | **1.35** | **3.00** |
| Mn (*M*1) | - | - | - | - | - | - | - | - | - | - | - | - | - | 0.23 |
| Ca (*M*1) | 4.07 | 4.65 | 3.41 | 3.86 | 4.71 | 3.30 | 3.64 | 4.35 | 3.47 | 3.95 | 3.27 | 4.46 | 5.49 | 4.68 |
| *REE* (*M*1) | 0.19 | 0.29 | 0.23 | 0.25 | 0.19 | 0.22 | 0.32 | 0.33 | 0.20 | 0.55 | 0.34 | 0.30 | 0.09 | 0.20 |
| Na (*M*1) | 1.74 | 1.06 | 2.36 | 1.90 | 1.11 | 2.48 | 2.04 | 1.33 | 2.33 | 1.50 | 2.39 | 1.24 | 0.42 | 0.67 |
| Y (*M*1) | - | - | - | - | - | - | - | - | - | - | - | - | - | 0.23 |
| **Sum** M1 | **6.00** | **6.00** | **6.00** | **6.00** | **6.00** | **6.00** | **6.00** | **6.00** | **6.00** | **6.00** | **6.00** | **6.00** | **6.00** | **6.00** |
| Na (*N*) | 14.04 | 14.38 | 12.69 | 10.91 | 10.64 | 13.09 | 10.72 | 14.00 | 12.21 | 13.58 | 6.24 | 13.43 | 11.62 | 10.59 |
| K(*N*) | 0.16 | 0.12 | 0.19 | 0.14 | 0.12 | 0.18 | 0.14 | 0.18 | 0.16 | 0.30 | 0.12 | 0.15 | 0.18 | 0.21 |
| Ba(*N*) | 0.04 | 0.04 | - | - | 0.03 | - | 0.15 | - | 0.05 | 0.12 | 0.04 | 0.03 | 0.04 | 0.03 |
| Sr (*N*) | 0.38 | 0.37 | 0.60 | 0.57 | 0.52 | 0.45 | 0.89 | 0.64 | 0.44 | 0.64 | 0.62 | 0.69 | 0.37 | 1.10 |
| **Sum *N*** | **14.61** | **14.91** | **13.48** | **11.62** | **11.30** | **13.72** | **11.90** | **14.82** | **12.86** | **14.64** | **7.02** | **14.30** | **12.20** | **11.92** |

*—core of zonal grain; **—edge of zonal grain.

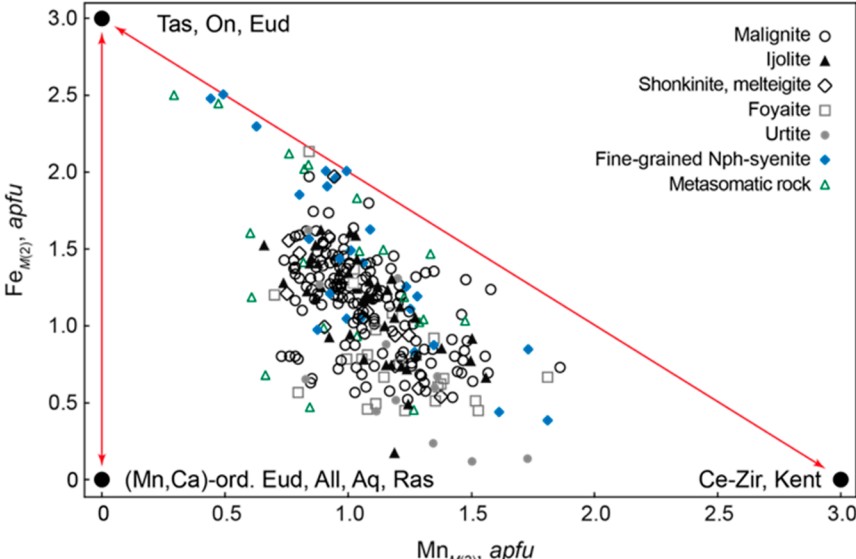

**Figure 16.** Manganese versus Fe on the M(2) site in EGM. For orientation, endmembers are shown. All—Alluaivite; Aq—Aqualite; Ce-Zir—Ce-Zirsilite; Eud—eudialyte; Kent—Kentbrooksite; On—Oneillite; Ras—Raslakite; Tas—Taseqite; (Mn,Ca)-ord. Eud— (Mn,Ca)-ordered eudialyte.

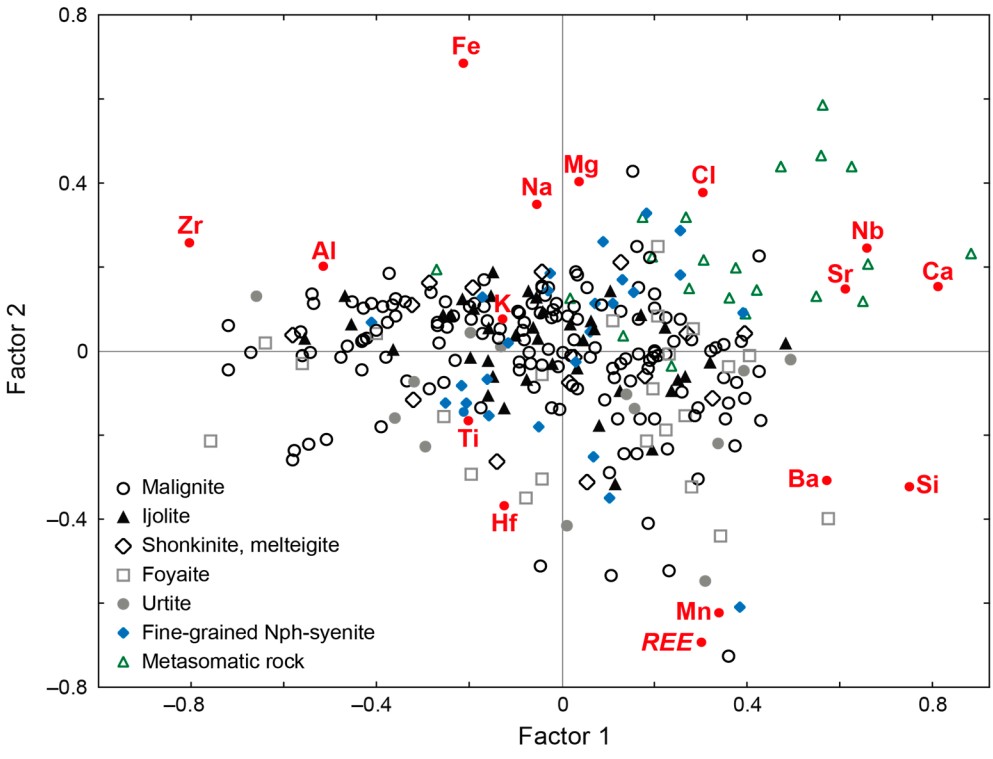

**Figure 17.** Results of factor analyses of EGM compositions. Var.—variables; Expl.var.—Explained variance; Prp.totl—proportion of total variance. Factor loadings > |0.5| are shown in bold.

| Var. | Factor loadings | | Var. | Factor loadings | |
|---|---|---|---|---|---|
| | Factor 1 | Factor 2 | | Factor 1 | Factor 2 |
| Na | 0.016 | 0.376 | Zr | **−0.791** | 0.302 |
| K | −0.144 | 0.063 | Hf | −0.160 | −0.386 |
| Ca | **0.831** | 0.105 | Nb | **0.679** | 0.206 |
| Sr | **0.620** | 0.092 | Ti | −0.203 | −0.104 |
| Ba | **0.553** | −0.351 | Si | **0.731** | −0.365 |
| REE | 0.261 | **−0.717** | Al | **−0.508** | 0.231 |
| Fe | −0.172 | **0.699** | Cl | 0.322 | 0.347 |
| Mn | 0.299 | **−0.648** | Expl.var. | 3.655 | 2.463 |
| Mg | 0.048 | 0.379 | Prp.totl | 0.215 | 0.145 |

## 5. Discussion

If apo-basalt metasomatic rocks are not taken into consideration, the Eudialyte Complex of the Lovozero Massif is composed of igneous rocks that can be formally subdivided into nepheline syenites (shonkinite, malignite and foyaite) and foidolites {melteigite, ijolite and urtite (Figure 2b)}. Based on the petrochemical (Figure 6), petrographic, and mineralogical data, these rocks should be divided into other groups. The first group includes hypersolvus meso- and melanocratic rocks (shonkinite, malignite, melteigite and ijolite) containing 30 or more modal % of mafic minerals. These rocks contain "streams" of dark-colored minerals (Figure 3a,b), have a trachytoid structure (Figure 3a), and are enriched with EGM. The $(Na_2O + K_2O)/Al_2O3$ ratio for these rocks ranges from 0.83 to 1.88 (Figure 7).

The second group includes subsolvus leucocratic rocks (foyaite, fine-grained nepheline syenite, urtite), where albite appears as primary magmatic mineral that present along with microcline-perthite (Figure 4a,c,e). This group is characterized by an elevated phosphorus content (Figure 6) responsible for the formation of various phosphates and silico-phosphates (Figure 4b). Also, the leucocratic rocks contain primary minerals of the lovozerite group (Figure 4d). Ratio $(Na_2O + K_2O)/Al_2O_3$ in these rocks ranges from 0.71 to 0.95 (Figure 7).

The relationships of rock-forming minerals of meso- and melanocratic rocks suggest that felsic minerals, namely alkali feldspar and nepheline, crystallized first. The earliest mineral was alkali feldspar (microcline) because the morphology and mutual orientation of its crystals (Figure 3a,b) indicate crystallization under conditions of free flow of magma [39,40]. In addition, there are inclusions of alkali feldspar in nepheline (Figures 3c and 18b), but not vice versa. It is convenient to consider the crystallization of felsic minerals using the "petrogeny's residua system" $NaAlSiO_4$-$KAlSiO_4$-$SiO_2$-$H_2O$ [41,42]. Phase equilibria in this system (Figure 18a) for "dry" conditions are defined in [43], and for $P_{H2O}$ = 1 kbar, phase equilibria can be found in [44].

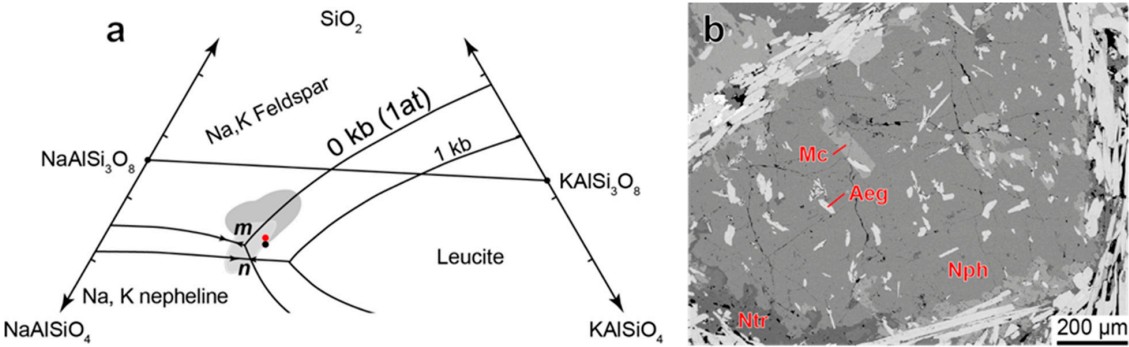

**Figure 18.** (**a**)—phase equilibrium diagram of the system $NaAlSiO_4$-$KAlSiO_4$-$SiO_2$-$H_2O$ for $P_{H2O}$ = 0 kbar and $P_{H2O}$ = 1 kbar (univariant lines after [43,44]); *m, n*—points of thermal minima for $P_{H2O}$ = 0 kbar (*m*) and $P_{H2O}$ = 1 kbar (*n*). Plot of the felsic normative compositions of the melanocratic (dark gray area) and leucocratic (gray area) rocks in comparison with the average composition of Eudialyte Complex (red point) and Layered Complex (black point), points are calculated after [14]. (**b**)—co-crystallization of nepheline, microcline and aegirine in malignite 147/221, BSE-image.

Figurative points corresponding to the rocks of the (meso-)melanocratic group form an area extending from the field of feldspar solid solution to nepheline–feldspar cotectic (Figure 18a). Compared with the average composition of the Eudialyte Complex (red point in Figure 18a), the normative composition of melanocratic rocks is enriched in feldspar. Alkaline feldspar crystallized first from the magmatic melt that spread laterally. Its fractionation quickly changed the composition of the melt towards cotectic and joint crystallization of nepheline-I and microcline began. The position of this cotectic was closer to the "dry" condition (*P* = 1 atm) due to the initially reducing nature of alkaline magma [7–9]. As a result, euhedral crystals of nepheline-I with small inclusions of microcline were formed (see Figures 3c and 18b). The formation of similar nepheline–feldspar (pseudoleucite-like) intergrowths in the process of cotectic crystallization is considered, for example, in work of Davidson [45]. Co-crystallization of nepheline-I and microcline is supported by the correlation of nepheline-I composition with Fe content in microcline (see Figure 11a). The constant admixture of ferric iron in microcline and nepheline indicates a high $Fe^{3+}$/$Fe^{2+}$ ratio already in the early stages of the rock crystallization. The oxidized state of iron is probably due to the "alkali-ferric-iron effect" [46] and this effect increases with decreasing temperature.

The crystallization temperature of nepheline-I is 500–775 °C ([31], Figure 9). The highest temperatures probably indicate liquidus state, whereas the lowest temperatures mark the transition to subsolidus hydrothermal conditions. The crystallization temperature of alkali feldspar from (meso-)melanocratic rocks is estimated below 700 °C (Figure 19a). However, since the rocks are hypersolvus, the temperature of feldspar crystallization cannot be lower than 650 ± 10 °C [47].

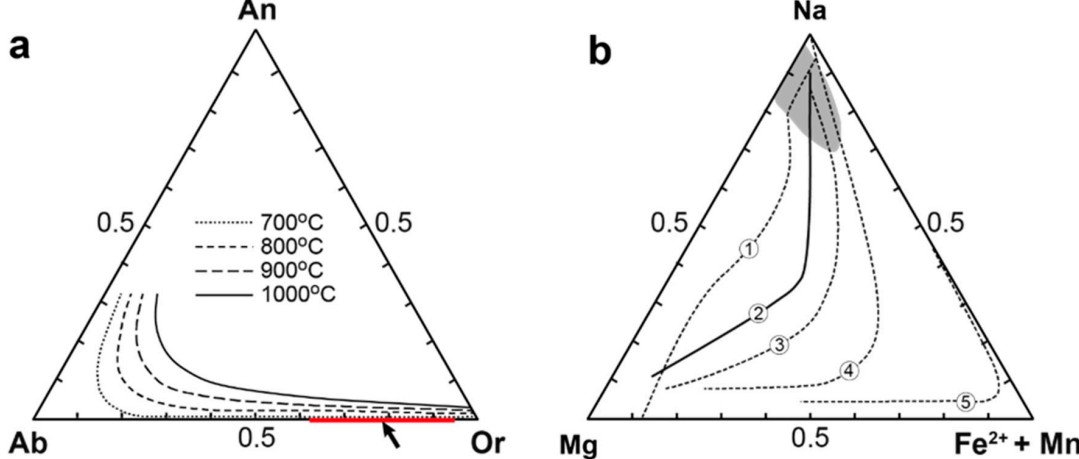

**Figure 19.** Compositions of felfspars and clinopyroxens from rocks of the Alluaiv site: (**a**)—total range of reintegrated feldspar compositions (red line) plotted on the temperature-dependent feldspar solvus, modified after [48] (from meso- and melanocratic rocks). (**b**)—total range of magmatic clinopyroxene compositions (gray field, generalized from Figure 11a). 1–5—clinopyroxene trends from other alkaline suites are shown for comparison: (1) Murun, Siberia [49], (2) Lovozero, Kola Peninsula [50], (3) Uganda [51], (4) South Qoroq [52], (5) Ilímaussaq [53].

After the crystallization of microcline (and nepheline-I), the magmatic melt lost its ability to flow freely. Microcline-perthite is the only mineral with deformed crystals (see Figure 3a), and this deformation occurred before its exsolution into microcline and albite. Long-prismatic aegirine-(augite) crystals are subparallel-oriented only in sections perpendicular to trachytoid plane. In sections that are parallel to the trachytoid plane, aegirine-(augite) crystals are irregularly oriented. Thus, after the crystallization of microcline (and nepheline-I), there was a transition from magmatic flow (suspension-like behavior) to "submagmatic flow" (flow with less than the critical amount of melt for suspension-like behavior) [40].

In the rocks of the Lovozero Eudialyte Complex, the calcium content is low; for (meso) melanocratic rocks, the average value is 1.81 wt. % CaO [14]. When alkali feldspar was fractionated, calcium accumulated in the melt. It is known that pure aegirine is not stable at magmatic temperatures, but even an insignificant increase in the calcium content served as a trigger for the aegirine crystallization [53,54]. The essence of this phenomenon is described in the work of Nolan [55] devoted to the albite-nepheline-aegirine-diopside system. This system includes two main components of clinopyroxene solid solutions (aegirineand diopside) and complements the region of residual petrogenetic system, which is poor in potassium and not saturated with silica. The addition of the diopside component to the clinopyroxene solution substantially changes the volume of the phase fields, and the field of clinopyroxene increases significantly. This effect can be traced to the composition of $Aeg_{50}Di_{50}$, after which the volume of the clinopyroxene solid solution remains almost unchanged.

As a result, after the formation of alkali feldspar crystals and almost simultaneously with nepheline-I, clinopyroxene began to crystallize. Initially, aegirine-(augite) crystallized by heterogeneous nucleation on the faces of growing nepheline crystals (but not feldspar ones). It can be assumed that the crystallization of nepheline "consumes less oxygen" (the ratio of the sum of cations to oxygen in nepheline is 3/4, and in the microcline is 2.5/4), and for the formation of aegirine-(augite) under reducing conditions, an oxygen donor is needed [6].

The next stage of the formation of (meso-)melanocratic rocks was a large-scale crystallization of aegirine-(augite) together with nepheline-II. Nepheline-II is enriched in Qtz endmember not because of the higher temperature of its crystallization, but due to the occurrence of $Fe^{3+}$ impurity during the substitution $\square_B + (Si^{4+} + Fe^{3+})_T \rightleftarrows K^+_B + 2Al^{3+}_T$. Therefore, nepheline-II cannot be used to estimate the temperature. Aegirine-(augite) from meso-melanocratic and leucocratic rocks are identical in

Mg-($Fe^{2+}$+Mn)-Na ratio (see Figure 12a,b). The sum $Fe^{2+}$ + Mn increases mainly due to manganese (Table 3), i.e., all iron during the clinopyroxene crystallization was in trivalent state. Magnesium (diopside endmember) enters clinopyroxene in the minimal amounts necessary only to stabilize it in magmatic conditions. The reason is the low calcium content in the melt. All clinopyroxenes are enriched with Na and $Fe^{3+}$, and located at the top of the fractionation trend (Figure 19b). Obviously, the rocks of the Eudialyte Complex are the most evolved (fractionated) among the alkaline rocks of the Lovozero Massif.

Composition of clinopyroxenes that form "streams" in meso-melanocratic rocks cannot be used to assess the oxygen fugacity due to the "alkali-ferric-iron effect" [46]; increased contents of Ti, Zr, Al and Mn in the clinopyroxenes indicate their rapid crystallization [56,57] and absence of the zirconium and titanium minerals at that time [58]. The antipathy of titanium and zirconium in clinopyroxenes (Figure 12a) has been established in other alkaline complexes, for example, Mont Saint-Hilare [59], Ilímaussaq [60]. Larsen [60] suggests that " ... these elements competed with varying success for a limited number of lattice sites".

The composition of amphiboles is completely correlated with the composition of coexisting clinopyroxenes (see Figure 15), i.e., crystallization switched from clinopyroxene to amphibole, probably because of rising $P_{H2O}$. The amphiboles do not contain zirconium but include titanium (due to crystal-chemical reasons), and the Na/Ca ratio in amphiboles is on average 24, which is twice as much as in clinopyroxenes. The early formation of EGM is incredible, as can be seen, for example, by comparing the compositions of the coexisting minerals: clinopyroxenes contain elevated concentrations of manganese, while the M2 position ($Fe^{2+}$, Mn, Mg) in the EGM constantly suffers from cation deficiency (see Figure 16). This means that the zirconium and calcium minerals, including EGM, crystallized together with amphiboles, but after clinopyroxenes. It disproves the hypothesis about early-magmatic formation of EGM [14,61].

In meso- and melanocratic rocks, the primary natrolite and sodalite crystallized simultaneously with amphiboles and EGM (see Figure 3c,d). Considering that sodic amphiboles are stable only at low temperatures (below 650 °C) and pressures [62,63] and dehydration temperature of natrolite is 350 °C [64], we can conclude that the primary natrolite crystallized almost simultaneously with alkaline amphiboles and EGM as a result of an increase in $P_{H2O}$. The binding of water in the composition of natrolite causes a decrease in the chlorine solubility and, consequently, the formation of sodalite. In this case, zonal natrolite–sodalite segregations appear, whose core consists of natrolite, the intermediate zone is composed of natrolite and sodalite, and the marginal zone consists of sodalite (Figure 3d). Thus, meso-melanocratic rocks were formed at temperatures ranging from 650–700 °C to about 350 °C. As the rocks crystallize in this temperature interval, a gradual transition from an almost anhydrous *HSFE*-, $Fe^{3+}$-, halogen-rich alkaline melt to the NaCl-rich water solution occurred.

Figurative points of leucocratic rocks (foyaite, urtite, fine-grained nepheline syenite) on the diagram of the "petrogeny's residua system", continue the field belonging to the points of meso-melanocratic rocks towards nepheline (see Figure 18a). The field of leucocratic rocks is located near the thermal minimum, but unlike meso-melanocratic rocks, this minimum is not "dry" (point m), but corresponds to $PH_2O$ = 1 bar (point n). Indeed, signs of simultaneous crystallization of microcline and nepheline are observed in foyaites and urtites, namely the inclusion of microcline in nepheline (see Figure 4a) and interrelation between microcline and nepheline compositions. The presence of primary albite in these rocks together with microcline-perthite indicates a high $PH_2O$ and low temperature ($\approx$550 °C) of mineral crystallization [65,66]. Amphiboles from meso-melanocratic rocks (enriched with Ca and Al) and leucocratic rocks (enriched with Si and Na) constitute the primary magmatic trend ([63], Figure 14). The Mn/Fe ratio in EGM is a fractionation indicator [38,67], and early-magmatic EGM are invariably dominated by Fe, whereas hydrothermal EGM can be virtually Fe-free and form pure Mn endmembers. EGM from leucocratic rocks are relatively rich in manganese.

The melt, which the leucocratic rocks crystallized from, was formed in the process of fractional crystallization of the melanocratic melt enriched in Fe and *HFSE*. Fractionation of the melanocratic

melt proceeded in the direction of enrichment with nepheline and a decrease in the aegirine content. A similar fractionation path occurs in the $Na_2O$-$Al_2O_3$-$Fe_2O_3$-$SiO_2$ system [68], where melt of the "ijolite" type (approximately 50% of aegirine) evolves towards "phonolitic eutectic" (approximately 10% of aegirine). Phonolite is similar in composition to a melt that is not saturated with silica at the minimum point of the "petrogeny's residua system" $NaAlSiO_4$-$KAlSiO_4$-$SiO_2$-$H_2O$ [41]. The residual nature of leucocratic rocks is also indicated by the association of accessory minerals [69]. Due to an excess of sodium silicate in these rocks, there are primary minerals of the lovozerite, lomonosovite, and murmanite groups. All these minerals have the highest possible percentage of sodium in the total cation number of the chemical formula.

The texture of leucocratic rocks (coarse-grained, fine-grained or porphyritic) depends on the proximity of the melt composition to nepheline–feldspar cotectic and the crystallization rate. The crystallization of subsolvus fine-grained nepheline syenite began with albite (see Figure 4a,b), followed by the almost simultaneous formation of microcline, nepheline, aegirine, and EGM. The simultaneous crystallization is indicated by correlations between compositions of coexisting minerals (Figure 11b). The leucocratic melt crystallized *in situ*, forming lenses (Figure 4g), layers, interlayers in the mass of melanocratic rocks. At slower cooling, the possibility of the formation of relatively large microcline and nepheline appeared (Figure 4a,c,e,f), and then a rapid and simultaneous crystallization of the main mass of the rock occurred.

Among the alkaline rocks of the Lovozero Massif, xenoliths of basic volcaniclastic rocks are widely distributed [18]. These rocks underwent metasomatic treatment of varying intensity and survived in the Eudialyte Complex both unchanged (see Figure 5a,b) and actually turned into nepheline syenites (see Figure 5c,d). In these rocks, there are all signs of a gradual increase in the intensity of alkaline metasomatism, including gradual transitions from unchanged basalt to fenite with relicts of augite surrounded by the rims of aegirine-(augite) (see Figure 13), and characteristic intergrowth of nepheline and magnesioarfvedsonite due to their simultaneous crystallization (see Figure 5d). This indicates the active supply of alkalis and the redistribution of calcium that localizes in fluorapatite and titanite. The duration of the thermal effect of alkaline melt on volcaniclastic rocks was small and, because of rapid cooling, feldspar remains homogeneous, but not completely exsolved into albite and ortoclase (Figure 8).

The wide variety of zirconium phases is due to the gradual increase in alkali concentration during fenitization. Early (relative to aegirine) crystallization of magnesioarfvedsonite indicates an elevated Si concentration and relatively low alkalinity, which also leads to the formation of zircon [59]. The relatively high fugacity of fluorine at the initial stage of fenitization [70] is also likely to favor the early formation of zircon, as demonstrated by the simultaneous formation of fluorapatite. The crystallization of aegirine leads to an increase in alkali content relative to silicon [71], which stabilizes parakeldyshite. EGM is formed later than parakeldyshite, at the final stage of fenitization.

## 6. Conclusions

1.  The Eudialyte Complex of the Lovozero Massif is indistinctly layered. Among the hypersolvus (meso)-melanocratic alkaline rocks (shonkinite, malignite, melteigite, ijolite) enriched with EGM, there are layers and lenses of subsolvus leucocratic rocks (foyaite, fine-grained foyaite and urtite) with phosphorus mineralization and primary lovozerite-group minerals.

2.  Leucocratic rocks were formed in the process of fractional crystallization of melanocratic melt enriched in Fe, *HFSE*, and halogens. The fractionation of the melanocratic melt proceeded in the direction of enrichment in nepheline and a decrease in the aegirine content. In a similar way, individual "rhythms" of the Layered Complex were probably differentiated.

3.  Hypersolvus meso-melanocratic rocks were formed in the temperature range from 650–700 °C to about 350 °C. As the rocks crystallized in this temperature range, a gradual transition from an almost anhydrous *HSFE*-, $Fe^{3+}$-, halogen-rich alkaline melt to the Na(Cl, F)-rich water solution occurred. The temperature of crystallization of subsolvus leucocratic rocks was about 550 °C;

4. If nepheline in alkaline rocks crystallizes simultaneously with aegirine, then it is enriched with Fe and Si according to the scheme $\square_B + (Si^{4+} + Fe^{3+})_T \rightleftarrows K^+{}_B + 2Al^{3+}{}_T$. The composition of such nepheline cannot be used for thermometry purposes;

5. Devonian volcaniclastic rocks played an important role in the giant eudialyte deposit formation. The relatively high fugacity of fluorine at first stage of the basalt fenitization causes formation of zircon in apo-basalt metasomatites. Following aegirine crystallization and the corresponding increase in Na/Si ratio led to parakeldyshite formation. At the final stage, EGM replaced parakeldyshite under the influence of Ca-rich solutions produced by the basalt fenitization.

**Supplementary Materials:** The following are available online at http://www.mdpi.com/2075-163X/9/10/581/s1, Table S1: parameters of chemical analyses; Table S2: modal composition of rock; Table S3: wet chemistry analyses of rock; Table S4: microprobe analyses of feldspar; Table S5: microprobe analyses of nepheline; Table S6: microprobe analyses of clinopyroxene; Table S7: microprobe analyses of amphibole; Table S8: microprobe analyses of eudialyte-group minerals.

**Author Contributions:** J.A.M. designed the experiments, participated in field works, performed statistical investigations and wrote the manuscript. A.O.K. participated in field works, performed geostatistical investigation, drew maps and reviewed the manuscript. Y.A.P. and A.V.B. took BSE images and performed electron microscope investigations. G.Y.I. and V.N.Y. conceived of the work, participated in field works, and reviewed the manuscript. All authors discussed the manuscript.

**Funding:** The Kola Science Center of Russian Academy of Sciences (0226-2019-0051), the Russian Science Foundation (16-17-10173), and the Presidium of the Russian Academy of Sciences (Program No I-48) funded this research.

**Acknowledgments:** We are grateful to anonymous reviewers from MDPI who helped us improve the presentation of our results.

**Conflicts of Interest:** The authors declare no conflict of interest.

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
