# Peer review of "Petrogenesis of the Eudialyte Complex of the Lovozero Alkaline Massif (Kola Peninsula, Russia)"

_minerals, doi:10.3390/min9100581_

Round 1

Reviewer 1 Report

Review of Ms# ‘minerals-576415-peer-review-v1’

"Petrogenesis of the Eudialyte Complex of the Lovozero Alkaline Massif (Kola Peninsula, Russia)" written by Mikhailova et al.

This study focuses its attention on the ‘Eudialiye complex’ in one of the most important alkaline intrusion, characterised mainly by nepheline syenites and other kind of rocks, with different mineralogical assemblages.

This topic is very interesting for scientific community and well presented.

Some corrections are necessary, they are listed below.

Some remarks:

The first important modification is to add this reference: Marks and Markl, 2017 regarding the definition of miaskitic, agpaitic and hyperagpaitic terms in the Introduction section.

Lines 119, 120, 184, 243, 283, 332, 366, 390, 412, 441, 442, 455, 606: It is correct ‘Alluaiv’ or ‘Alluaive’. Please revise the text.

Line 158: Move this table as Supplementary; but you can leave the mineral abbreviation in the text.

Lines 346, 352 and 353: Replace ‘Mi’ with ‘Or’ it is better, because you use in the table ‘Or’. In addition ‘Mi’ is unusual.

Line 373: modify ‘minal’.

Line 444: (Fe2+ + Mn) or (Fe2+ + Mn – Na), Fe2+ or Fe 3+ or Fe tot?

For me it is correct in this way (Fetot+ Mn – Na).

Section 5. Discussion

It is very interesting, but I suggest you to read also these two articles Melluso et al., 2017 and Guarino et al., 2019, and add them in the reference list.

More attention is due when the authors talk about the mineral crystallization sequence.

For example I believe that aegirine crystallizes together nepheline.

Tables and supplementary tables:

In all tables, list the cationic elements in the same order of the major oxides. Add the cationic calculation also in the supplementary materials.

In table 3, how you have calculated the Nph, Ks, and Qtz values? Are they mol.% end members?

Several eudialyte analyses have a very low ‘total’ of oxides sum, see supplementary table. How you explain this? (To lower than 93 wt.%). Are these analyses stechiometric?

Delete the column of THO2, it is always 0.

In the supplementary file of clinopyroxene there are four analyses with MgO high than 10 wt.% and lower Na2O. Describe it. Also the clinopyroxene have analyses with ‘total’ of oxides sum is lower than 96 wt.%. Are these analyses stechiometric?

In the supplementary file of amphiboles, they have analyses with ‘total’ of oxides sum is lower than 94 wt.%. Are these analyses stechiometric?

References:

Guarino V., de’ Gennaro R., Melluso L., Ruberti E., Azzone R.G., 2019. The transition from miaskitic to agpaitic rocks as marked by the accessory mineral assemblages, in the Passa Quatro alkaline complex (southeastern Brazil). The Canadian Mineralogist, 57, 1–23, doi: 10.3749/canmin.1800073. Melluso L., Guarino V., Lustrino M., Morra V., de’ Gennaro R., 2017. The REE- and HFSE-bearing phases in the Itatiaia alkaline complex (Brazil), and geochemical evolution of feldspar-rich felsic melts. Mineralogical Magazine 81(2), 217–250, doi: 10.1180/minmag.2016.080. Marks, M.A.W. & Markl, G., 2017. A global review of agpaitic rocks. Earth-Science Reviews 173, 229–258.

Author Response

See an attached file.

Reviewer 2 Report

The paper looks at an interesting group of rocks with unusual mineralogies: the Eudialyte Complex of the Lovozero Alkaline Massif. This is worthy of scientific publication. However, there paper is not ready for submission to a journal. 

I understand that writing in English for non-native speakers is difficult. This means extra care must be made in ensuring the written message is clear. I cannot clearly evaluate the scientific merits of the paper due to the lack on clarity in regards to grammar, spelling, word choice, and syntax.

This paper must undergo a focused edit to improve the clarity of writing. One suggestion is to avoid the use of archaic rock names that are not exactly IUGS compliant: malignite, urtite, shonkinite, etc. Instead: aegerine nepheline syenite, nephelite, olivine augite nepheline syenite, etc.

A few specific points: the word 'illegible' shows up in a line 14 and 'non-legible' shows up in line 724. Do they mean 'cryptic' layering?

Line 28: Devonion 'volcaniclastic' rocks...

Rock names are collective nouns: 'There is syenite', not 'There are syenites'

Why italics on line 16 and 18 and others?

Reference #3 and 4 would not be examples of 'recent reviews' of these rare metal deposits.

Please overhaul/re-write the paper in proper English in order for the science to be effectively evaluated by peer review at a later date.

Sincerely,

Author Response

See an attached file.
